# Dust aerosol from the Aralkum Desert influences the radiation budget and atmospheric dynamics of Central Asia

Jamie R. Banks[1], Bernd Heinold[1], and Kerstin Schepanski[2]

[1]Leibniz Institute for Tropospheric Research (TROPOS), Leipzig, Germany.
[2]Institute of Meteorology, Freie Universität Berlin, Berlin, Germany.

**Correspondence:** Jamie R. Banks (banks@tropos.de)

**Abstract.**

The Aralkum is a new desert created by the desiccation of the Aral Sea, and is an efficient source of dust aerosol which perturbs the regional Central Asian radiation balance. COSMO-MUSCAT model simulations are used to quantify the direct radiative effects (DREs) of Aralkum dust, and investigate the associated perturbations to the atmospheric environment. Considering scenarios of 'Past' and 'Present' defined by differences in surface water coverage, it is found that in the Present scenario the simulated yearly mean net surface DRE across the Aralkum is -1.34 W m$^{-2}$ with a standard deviation ($\pm$) of 6.19 W m$^{-2}$, of which -0.15$\pm$1.19 W m$^{-2}$ comes from dust emitted by the Aralkum. In the atmosphere the yearly mean DRE is -0.62$\pm$2.91 W m$^{-2}$, -0.05$\pm$0.51 W m$^{-2}$ from Aralkum dust: on the yearly timescale Aralkum dust is cooling both at the surface and in the atmosphere, due to its optically scattering properties. The daytime surface cooling effect (solar zenith angle $\lesssim$70-80°) outweighs the nighttime heating effect and the atmospheric daytime (solar zenith angle $\lesssim$60-70°) heating and nighttime cooling effects. Instantaneous Aralkum dust DREs contribute up to -116 W m$^{-2}$ of surface cooling and +54 W m$^{-2}$ of atmospheric heating. Aralkum dust perturbs the surface pressure in the vicinity of the Aralkum by up to +0.76 Pa on the monthly timescale, implying a strengthening of the Siberian High in winter and a weakening of the Central Asian Heat Low in summer. These results highlight the impacts of anthropogenic lakebed dust on regional atmospheric environments.

## 1 Introduction

The formation of the Aralkum (the Aral Desert, Breckle and Wucherer (2012)) due to the desiccation of the Aral Sea over the last 60 years (Micklin, 2010) has generated a new source of atmospheric dust aerosol in Central Asia (e.g. Groll et al., 2013; Nobakht et al., 2021, and others). The Aral Sea is supplied by water from the Amu Darya and Syr Darya rivers (originating in the Pamir and Tian Shan mountains), however since the 1960s the water from these rivers has been exploited more extensively for irrigation, which has resulted in a dramatic reduction in the outflow to the Aral Sea. Approximately 60,000 km$^2$ or ~90% of the former lakebed has been exposed by this desiccation, revealing an erodible surface efficient at emitting dust into the atmosphere given favourable wind conditions. Among other impacts, this has had particularly negative consequences for air quality and human health in the region, exacerbated by the pesticide and fertiliser chemicals in the Aralkum's soils associated with the irrigation practices which created its desert (e.g. O'Hara et al., 2000; Wiggs et al., 2003, and others). Given that the

formation of the Aralkum is a manmade disaster, dust from the Aralkum may therefore be regarded as being anthropogenic in origin.

It is well known that an environmental impact of dust aerosol is on the Earth's radiation balance, in terms of both solar and thermal radiation (e.g. Hsu et al., 2000; Banks et al., 2014; Alamirew et al., 2018, and others), and so it is to be expected that dust from the Aralkum would add an extra perturbation to the Earth's radiation environment. With respect to shortwave (SW)
solar radiation, in its optical properties dust is highly scattering but also partially absorbing (e.g. Haywood et al., 2003; Ryder et al., 2013, and others), with the effect that during daylight hours lofted dust aerosol tends to cool the surface underneath it, but warms the atmospheric layer in which it is located. The longwave (LW) thermal direct radiative effect (DRE) of dust is more complex, and relates not only to the optical properties of the dust but also to the atmospheric temperature and moisture profiles, and to the altitude and vertical distribution of the dust layer (e.g. Haywood et al., 2005; Brindley and Russell, 2009,
and others). In hot and cloud-free desert environments, in terms of LW radiation dust is likely to have a warming effect on the surface beneath it, since it traps LW radiation being emitted by the surface, as well as absorbing incoming SW radiation from above. The behaviour of the LW DRE is more complicated at night, during winter, and over more diverse non-desert environments such as are also present in Central Asia. The net (SW+LW) radiative effect of dust can therefore be cooling or warming, depending on the altitude of the dust within the atmosphere (Meloni et al., 2018), the time-of-day (Osipov et al., 2015)
and the season, the surface albedo (Tegen et al., 2010; Li and Sokolik, 2018), and the precise mineralogy, optical properties (Di Biagio et al., 2020; Adebiyi et al., 2023) and the size (Kok et al., 2017) of the dust.

This paper builds on the Central Asian dust modelling study presented in Banks et al. (2022), henceforth denoted BHS22, to explore the consequences of Aralkum dust for radiative effects over Central Asia. BHS22 presented regional model simulations of dust emissions and transport from the deserts of Central Asia and specifically from the Aralkum under three scenarios of Aral
Sea water coverage, considering the 'Past' (end of 20th century), the 'Present' (beginning of 21st century), and an Aralkum lakebed-only scenario as a sensitivity study. The concept of the Present scenario is to produce as accurately as possible a current representation of the dust emissions and transport from all the Central Asian dust sources with the inclusion of the Aralkum. Meanwhile the concept of the Past scenario is to depict the state of the dust environment as it was three to four decades ago (including dust from the pre-existing deserts of the Karakum and the Kyzylkum in Turkmenistan and Uzbekistan), when the
Aral Sea was much more extensive than it is today. The Aralkum-only scenario is a less physical case, an idealised scenario which excludes the other Central Asian deserts so as to investigate the atmospheric dust concentrations of dust exclusively from the Aralkum. Over the course of a 1-year case study simulation period, it was found that there has been a near-doubling in dust emissions from the Aralkum region (43-47°N, 58-62°E, including non-lakebed surfaces) between the Past and the Present due to the expansion of the Aralkum. In terms of air quality, settlements as far as ∼600-800 km downwind to the
east of the Aralkum may be badly affected by its dust. It is also clear that there is a high degree of interannual variability in the directions of the dust-emitting winds over the Aralkum, such that in other years other directions are more susceptible to Aralkum dust. Problematically for observations of the Aralkum's dust, approximately two-thirds of the Aralkum's dust emissions are simulated to occur under cloudy skies, and hence would be impossible to observe with traditional passive remote sensing techniques. Open questions remain, however, as to the dust radiative effects: what may be the consequences of the

changes in Aralkum dust activity for the radiative effects of dust over the region? What are the patterns of radiative cooling and warming on the surface and atmosphere of the Aralkum region, can these be quantified, and how have these changed as the Aral Sea has shrunk and the Aralkum has expanded? Moreover, have these radiative effects had consequences for the wider atmospheric environment of the region?

In order to explore these questions as to the radiative effects of Aralkum dust, this paper takes the following structure: in Section 2 we describe the COSMO-MUSCAT model setup used to simulate regional dust emissions and transport, and dust's influence on the regional SW and LW radiative fluxes; in Section 3 we explore the behaviour and patterns of dust radiative effects over the Aralkum; in Section 4 we consider the changes in the radiative effects due to growing dust emissions from the Aralkum (i.e. the difference between 'Past' and 'Present'); and finally in Section 5 we consider perturbations to the atmospheric state due to the presence of Aralkum dust in the atmosphere.

## 2 The COSMO-MUSCAT model and its treatment of dust radiative effects

### 2.1 COSMO-MUSCAT, the 'Dustbelt' domain, and dust emission

The regional aerosol model COSMO-MUSCAT (COSMO: COnsortium for Small-scale MOdelling; MUSCAT: MUltiScale Chemistry Aerosol Transport Model) is based on version 5.05 of the atmospheric model COSMO (Schättler et al., 2014), which is coupled online with the chemistry tracer transport model MUSCAT (Wolke et al., 2012). Mineral dust is an aerosol species which may be modelled using COSMO-MUSCAT, simulating dust transport processes including emission and dry and wet deposition, thereby providing 3D dust atmospheric concentrations (Heinold et al., 2011). Dust radiative effects are also simulated (see Section 2.2).

The Central Asian domain is contained within the 'Dustbelt' ('DUBLT') modelling domain (BHS22), which spans the deserts of North Africa, the Middle East, and Central Asia (Fig. 1). The DUBLT domain is projected onto a rectangular rotated pole grid with an origin at 20°N, 45°E (in Saudi Arabia), and with a domain size of 160 latitudinal grid cells by 360 longitudinal grid cells. The grid spacing is 0.25° (28 km), which produces a domain that extends as far north as ~50°N in Kazakhstan and as far east as ~84°E in China. For the purposes of the current study, focusing on the radiative effects of dust from the Aralkum and its neighbouring deserts, the Central Asian component of the model domain is considered to be bounded by 35-49°N, 48-72°E.

The parameterisation of dust emission follows the saltation bombardment scheme of Marticorena and Bergametti (1995), a formulation which calculates first the horizontal saltation flux $G$, for given soil properties and wind speeds. $G$ has a linear relationship with the vertical emission flux $F$. The sandblasting efficiency $\alpha$, dependent on the soil size distribution, is the coefficient of proportionality between $F$ and $G$. Soil properties are described by version 2.0 of the SoilGrids250m dataset (Hengl et al., 2017; Poggio et al., 2021), although missing data within the former Aral Sea basin are filled in with measurements made by Argaman et al. (2006) of soils in the southern Aralkum. The roughness length $z_0$ is described by retrievals from ASCAT and PARASOL data performed by Prigent et al. (2012); missing $z_0$ data within the Aralkum are filled in using the minimum value from the vicinity of the basin, which for the Aralkum is 0.02 cm. Dust emission is suppressed in the presence

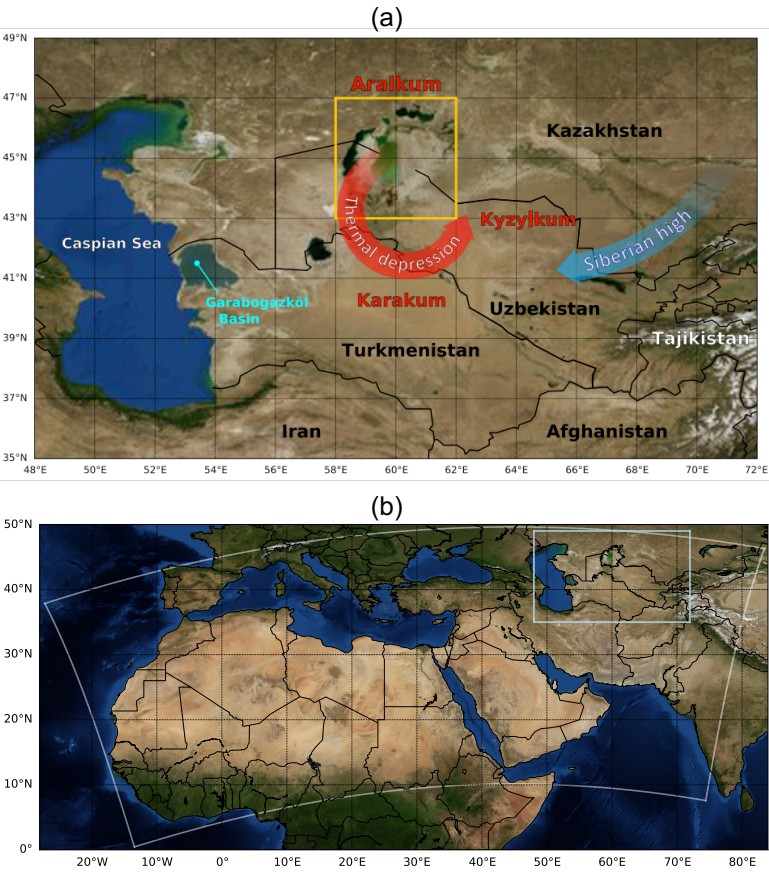

**Figure 1.** Map of Central Asia (panel (a)), marking its countries, deserts and water bodies, overlaid on the NASA 'Blue Marble' image. The orange box encompasses the Aralkum. Relevant major Central Asian weather patterns are named (as described on p. 14 of Global Facility for Disaster Risk Reduction and Recovery (2019)), with arrows indicating their flow regimes and colours indicative of their temperature regimes. In panel (b), the white box outlines the edge of the 'DUBLT' modelling domain, while the light blue box denotes Central Asia.

of vegetation and snow (both derived from Aqua MODIS retrievals (Didan, 2015; Hall and Riggs, 2016)), such that:

$$F = \alpha * (1 - A_{\mathrm{snow}}) * A_{\mathrm{eff}} * A_{\mathrm{land}} * G, \tag{1}$$

where $A_{\mathrm{eff}}$ is the 'effective' bare and unvegetated surface area fraction, and $A_{\mathrm{snow}}$ and $A_{\mathrm{land}}$ are the snow and land area fractions.

The DUBLT modelling system has been developed in order to evaluate the effects of changes in surface water coverage on dust emissions and transport (BHS22). It does this by making use of the Global Surface Water dataset (Pekel et al., 2016), which makes use of aggregated Landsat data during the periods 1984-1999 (interpreted within DUBLT as the 'Past') and 2000-2015 (the 'Present'). The dataset is publicly available at https://global-surface-water.appspot.com/ (last accessed 11th

July 2023), and has subsequently been updated to extend to 2020. For more extensive information on the DUBLT modelling system and its output in terms of dust emission and transport, see BHS22.

## 2.2 Model configuration of dust and radiation

Within COSMO-MUSCAT dust is considered to be a passive tracer, with five independent size bins ($j$) with radii bounded by 0.1 and 24.0 $\mu$m. The effective radii of these bins have values $r_{\text{eff}} = (0.169, 0.501, 1.514, 4.570, 13.80)\,\mu$m. The dust shape assumption is spherical, and the particles are assumed to have the density of quartz, 2.65 g cm$^{-3}$. Dust radiative feedbacks are switched on so as to include the effects of dust on the atmospheric dynamics (Helmert et al., 2007). With respect to the calculation of the dust aerosol optical depth (DOD) at 550 nm, the dust refractive indices are defined as described by Sinyuk et al. (2003). Since the simulated dust particles are spherical, Mie Theory (Mie, 1908) can be used to calculate the extinction efficiencies $Q_{\text{ext}}$: for the five size bins $Q_{\text{ext}} = (1.677, 3.179, 2.356, 2.144, 2.071)$. The DODs are then obtained by multiplying the height-resolved ($k$) dust concentrations $c_{\text{dust}}(j, k)$ by $Q_{\text{ext}}(j)$ and integrating over the atmospheric column:

$$DOD = \sum_j \sum_k \left( \frac{3}{4} \frac{Q_{\text{ext}}(j)}{r_{\text{eff}}(j)\rho_{\text{p}}} c_{\text{dust}}(j,k)\Delta z(k) \right), \tag{2}$$

where $\rho_{\text{p}}$ is the particle density and $\Delta z(k)$ is the thickness of each atmospheric layer.

Atmospheric radiation fluxes are modelled via eight spectral bands (Helmert et al., 2007): three of these bands comprise the SW broadband which covers the spectral range of 0.25-4.64 $\mu$m, while the other five spectral bands comprise the LW, which covers the range 4.64-104.5 $\mu$m. Radiative transfer is computed using the radiation scheme described by Ritter and Geleyn (1992), deploying a $\delta$-two-stream radiative transfer solver approach. The impact of dust on the radiation fluxes across the SW and LW bands is simulated using refractive indices collected and described by Helmert et al. (2007).

Dust direct radiative effects (DRE) in the SW and LW are computed with reference to the DUBLT_NODUST scenario, which is a COSMO-MUSCAT run without any dust emissions and hence with atmospheric dust concentrations set to zero. Within the model the convention is that radiative fluxes are positive in the downwards direction, and hence positive net surface (SFC) fluxes $SW_{\text{SFC}}$ and $LW_{\text{SFC}}$ imply a net warming of the surface. Similarly, positive values of the top-of-the-atmosphere (TOA) net fluxes imply a net warming of the Earth-atmosphere system. COSMO-MUSCAT provides both SFC and TOA net fluxes as model output. In the SW, the surface DRE ($DRE_{\text{SW,SFC}}$) is therefore defined as:

$$DRE_{\text{SW,SFC}} = SW_{\text{SFC,DUST}} - SW_{\text{SFC,NODUST}}. \tag{3}$$

The same equation holds for the LW and for the TOA fluxes. From the difference between the TOA and the SFC DREs, it is straightforward to isolate the atmosphere-only (ATM) DREs:

$$DRE_{\text{SW,ATM}} = DRE_{\text{SW,TOA}} - DRE_{\text{SW,SFC}}. \tag{4}$$

In order to exclude the substantial confounding effects of cloud and to identify the dust-only radiative effects, DRE calculations are only performed up to a fairly stringent maximum of 1% cloud cover in both of the DUST and NODUST simulations. Throughout this paper it is important to recall that at a longitude of $\sim$60°E, the local time over the Aralkum is approximately four hours ahead of UTC.

When considering dust radiative effects on the surface and on the atmosphere, the surface albedo is essential background information, maps of which are presented for the first days of six months during the DUBLT simulation year in Fig. 2. MODIS

(Moderate Resolution Imaging Spectroradiometer) albedo retrievals are used for this purpose within the COSMO-MUSCAT simulations, albeit modified by surface snow and ice generated within the COSMO simulations. Snow and ice are particularly noticeable at the beginning of April, December and February over the north of the domain in Kazakhstan, as well as over the Tian Shan and Pamir mountains, and also over the modelled lake surface of the Aral Sea especially in February. In contrast during the late spring and the summer and autumn months, the albedo of the lake of the Aral Sea is distinctly lower than that of its surroundings. It is therefore to be expected that the simulated radiative effects of dust directly over the Aral Sea will be strongly dependent on the season in which it is emitted, and on the simulated freezing state of the lake.

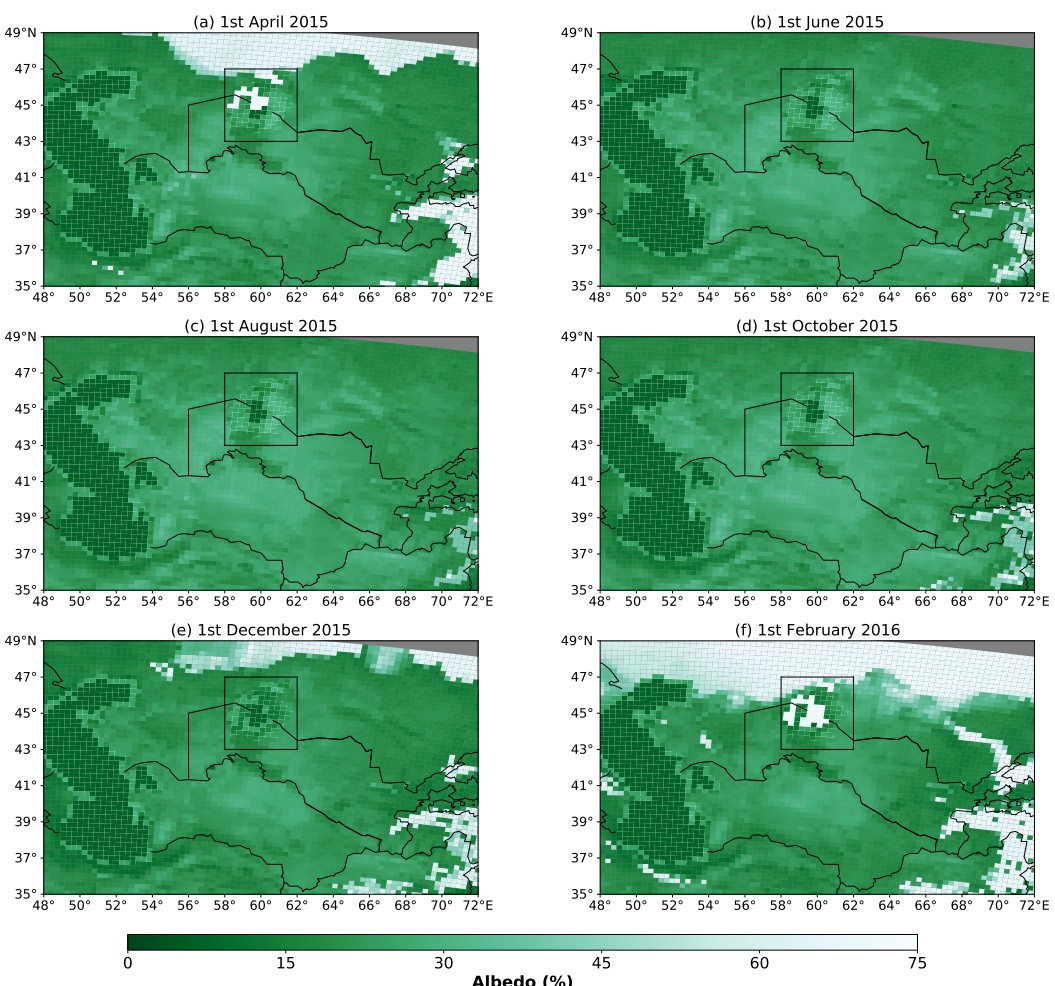

**Figure 2.** Monthly maps of surface albedo (%) over Central Asia, from 0900 UTC on 1st of each month. Grey areas to the north lie outside of the DUBLT domain. Within the Aralkum (43-47°N, 58-62°E) the mean albedo is (a) 29.5%, (b) 22.7%, (c) 23.5%, (d) 23.5%, (e) 19.1%, and (f) 34.3%.

Within COSMO-MUSCAT there are multiple possibilities as to the assumption of the dust optical properties. Within this paper, we make use of a dust type derived by Helmert et al. (2007) for use within MUSCAT, which is considered to be a 'reflecting' (i.e. scattering) dust type. This reflecting dust is aggregated from four dust optical properties databases within wavelength ranges from 0.25-0.44 $\mu$m (Sinyuk et al., 2003), 0.44-1.02 $\mu$m (Dubovik et al., 2002), 1.02-2.52 $\mu$m (Sokolik and Toon, 1999), and 2.52-35.19 $\mu$m (Volz, 1973). At IR wavelengths > 35.19 $\mu$m, the refractive indices are set to be constant at the 35.19 $\mu$m values. These dust optical properties were derived from remote sensing observations, in-situ retrievals at Cape Verde, laboratory measurements, and bulk sampling, respectively. The reflecting dust is currently regarded as the default dust type in COSMO-MUSCAT, and was also the dust type assumed in BHS22. Moreover, given the particular prevalence of salt dust in the Central Asian region originating from dry lakebeds such as the Aralkum (Orlovsky et al., 2005; Argaman et al., 2006; Hofer et al., 2020; Zhang et al., 2020; Xi, 2023), it is reasonable to make the assumption that the dust in this region would be a more reflecting dust type. Helmert et al. (2007) also analysed a more absorbing dust type, derived from the laboratory measurements made by Sokolik and Toon (1999), assuming an internal mixture of 98% kaolinite and 2% hematite.

Within the context of this study, there are four dust model scenarios: 1) the baseline DUBLT scenario, representative of 'Present' surface water coverage and hence Present dust emissions (also known as 'DUBLT_PRESENT'); 2) DUBLT_PAST, representative of Past dust emissions; 3) DUBLT_NODUST, a simulation run without dust emissions so as to provide information about the atmospheric state in the absence of dust; and 4) DUBLT_ABS, an alternative version of the Present scenario using the 'absorbing' dust type, a sensitivity study used in Section 4.2. The first two scenarios were presented and described in BHS22. The simulation period is the year from 30th March 2015 - 29th March 2016, with a two-week model spinup period beforehand to initialise the dust concentrations.

## 2.3 Quantification of the uncertainties in the COSMO-MUSCAT radiative fluxes

In order to ascertain the uncertainties in the simulated radiative fluxes over the Aralkum (before we can even consider the effects of dust), Fig. 3 compares the DUBLT simulated TOA fluxes with those retrieved from the CERES (Clouds and the Earth's Radiant Energy System) instrument onboard NASA's Aqua and Terra satellites (Wielicki et al., 1996; Loeb et al., 2001; Doelling et al., 2013). These CERES TOA flux data are the Level 3 SSF1deg (Single Scanner Footprint) Aqua data, which are daily mean values spatially averaged onto a uniform 1° grid. The CERES SW band is in the spectral range 0.3-5 $\mu$m, and the LW band is from 5-200 $\mu$m (Loeb et al., 2016): note that these are different from the COSMO-MUSCAT bands of 0.25-4.64 and 4.64-104.5 $\mu$m as mentioned previously in Section 2.2. Given the well-understood seasonal cycle in the atmospheric fluxes, it is to be expected that the overall correlations between the simulated and the retrieved fluxes are very close to 1, slightly better simulated in the SW than the LW. The net TOA fluxes switch from positive to negative approximately at the beginning of September, and return to being positive later in March. In general the DUBLT fluxes are biased high, up to +8.5 W m$^{-2}$ in the SW, +1.9 W m$^{-2}$ in the LW and +10.4 W m$^{-2}$ for the net fluxes. Most importantly, considering the root-mean-square differences (RMSDs) as an estimate of the uncertainties on the DUBLT simulated fluxes, the uncertainty on the SW fluxes would therefore be ±9.7 W m$^{-2}$ while that for the LW is lower at ±5.6 W m$^{-2}$. Cumulatively this gives an uncertainty on

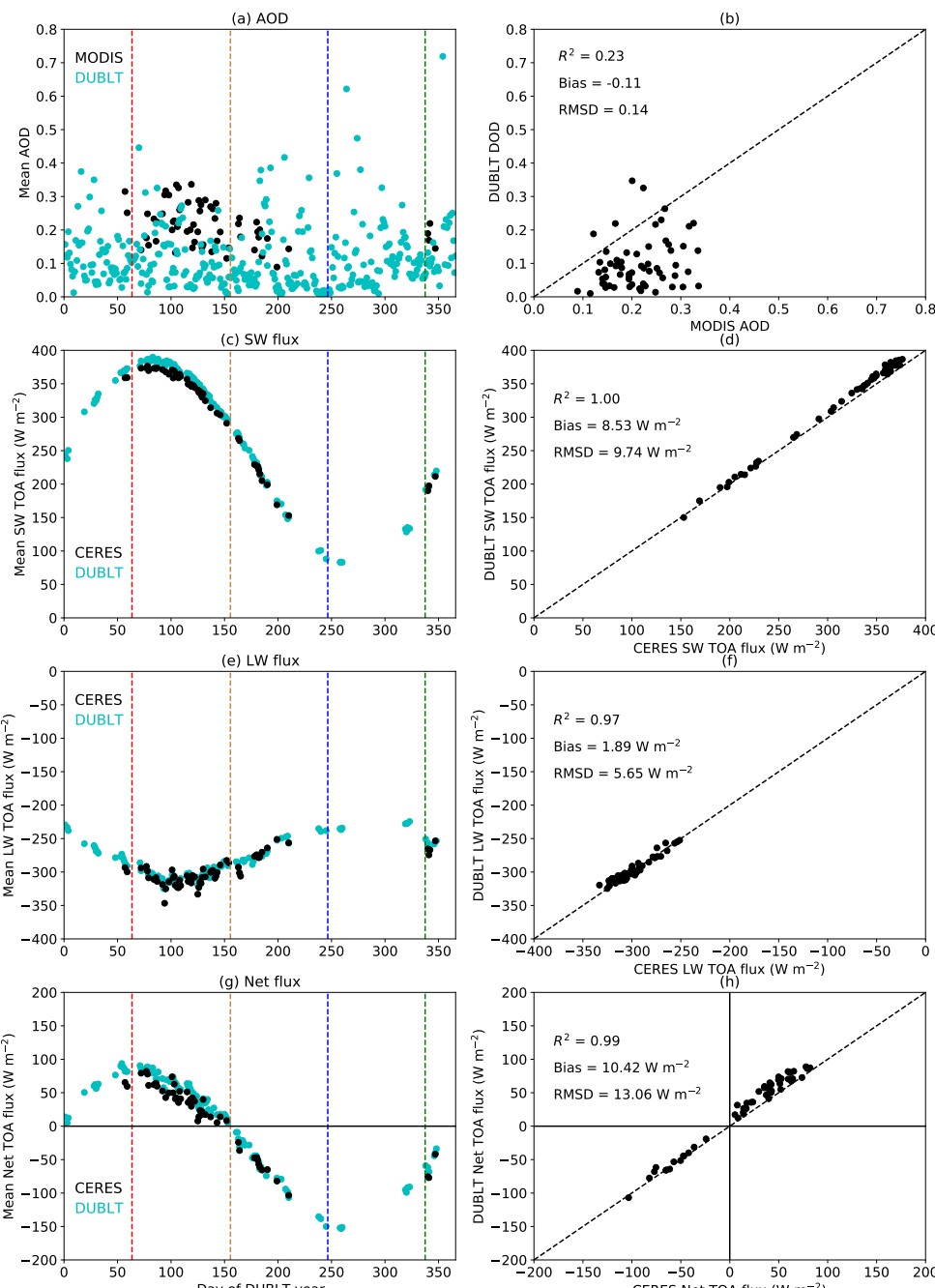

**Figure 3.** Timeseries plots (left column) of daily mean DUBLT DODs and Aqua MODIS AODs, DUBLT and Aqua-CERES SW, LW, and net fluxes, averaged over the Aralkum (see the red box in Fig. 1), along with the associated scatterplots (right column). For the DODs and DREs only grid cells with a cloud cover $< 1\%$ go into the data subset for each daily point in the timeseries. Of the 201 grid cells in the Aralkum, more than 10% of the grid cells must fulfil this criterion in order for the point to go into the timeseries. A total of 53 days go into this co-located data subset. The MODIS AODs are filtered by the retrieved Ångström coefficient as being $< 0.6$ (e.g. Dubovik et al., 2002; Banks et al., 2013), as an attempt to discriminate primarily dust aerosols. The RMSD is the root-mean-square difference. The dashed vertical lines denote the change of the seasons: the red line is the start of summer (1st June), the brown line the start of autumn, the blue line the start of winter, and the green line the start of the second spring (1st March 2016).

the net fluxes of $\pm 13.1\,\mathrm{W\,m^{-2}}$. For further information as to the performance of the DUBLT modelling scheme in simulating regional dust loadings, see Section 4.1 in BHS22.

## 3 Overview of the dust direct radiative effects in the Aralkum region during the DUBLT year

For an overview of the Aralkum's dust activity over the year, Fig. 4 depicts the cloud-screened timeseries of the Aralkum's dust emissions, DODs and dust DREs, for the approximate midday and midnight timeslots (0900 and 2100 UTC, plotted separately). The data are cloud-screened so as only to consider those timeslots when DRE calculations are successfully performed. The timeseries of the emissions and DODs (Fig. 4, panels (a-d)) show the extent to which dust activity is event-based, with high DODs occurring only occasionally: notable daytime peaks occur in winter and March 2016, while notable nighttime peaks are in autumn and later in the winter. Due to strict cloud-screening there are gaps in the timeseries (especially in winter), and so a number of dust events are not depicted in this timeseries as a result of being coincident with cloud cover. The co-location of cloud coverage and dust emissions was discussed in detail in BHS22, which found that over two-thirds of the simulated yearly emissions occurred under extensive cloud cover, including the two days in March 2016 (15th and 16th) which had the highest daily emissions. Within the two emission timeseries (Fig. 4, panels (a) and (b)) it seems to appear that emission occurs somewhat more frequently in the morning: for cloud-free grid cells (cloud cover $< 1\%$) the yearly total of the 0300-0900 UTC emissions is 1.23 Tg, while for 1500-2100 UTC it is 0.56 Tg. However under all-sky conditions (not plotted) these emission values show more equivalence at 7.51 and 5.22 Tg, showing that there is a larger fraction of the evening emissions occurring under cloudy skies: 10.8% of the evening emissions occur under clear-skies as opposed to 16.4% in the morning.

Due to the lag in dust accumulation in the atmosphere after emission, and due to external dust sources, this discrepancy between the morning and evening emissions is not replicated in the DODs. The mean values of the points in the cloud-screened timeseries are 0.104 at 0900 UTC and 0.093 at 2100 UTC. As with the emissions, the day with the peak morning DODs is 18th December 2015 (see also Figure 12(c-d) in BHS22 for the DOD case study of this day). This is also the day with the largest Aralkum mean values of SW SFC cooling and LW SFC heating, at -71.1 and +32.8 $\mathrm{W\,m^{-2}}$ respectively in the Present scenario. In the atmosphere the SW and LW DREs almost exactly balance each other out on this day, with a SW DRE of +25.0 $\mathrm{W\,m^{-2}}$ balanced against a LW DRE of -25.7 $\mathrm{W\,m^{-2}}$, showing how the most prominent dust events do not necessarily lead to the clearest radiative heating or cooling situations. In general, however, the net effect of dust over the Aralkum during the daytime is to cool the surface and warm the atmosphere, DREs which are dominated by the SW effect. Meanwhile at night there is of course no instantaneous SW effect, so the nighttime SFC and ATM DREs are governed solely by the LW radiative heating and cooling, respectively.

Considering the explicit effects of dust on the atmospheric radiation environment, it is instructive and intuitive to consider first an instantaneous case study, before aggregating the simulation output to explore the relationships between variables. As indicated by the dust emissions in Fig. 4, dust activity tends to occur irregularly on an event basis, such that the yearly averages may not always be a clear guide as to the patterns that occur during a substantial dust outbreak. Moreover, by mapping consecutive timeslots it is possible to observe the progression of the radiative effects of an individual dust event with respect

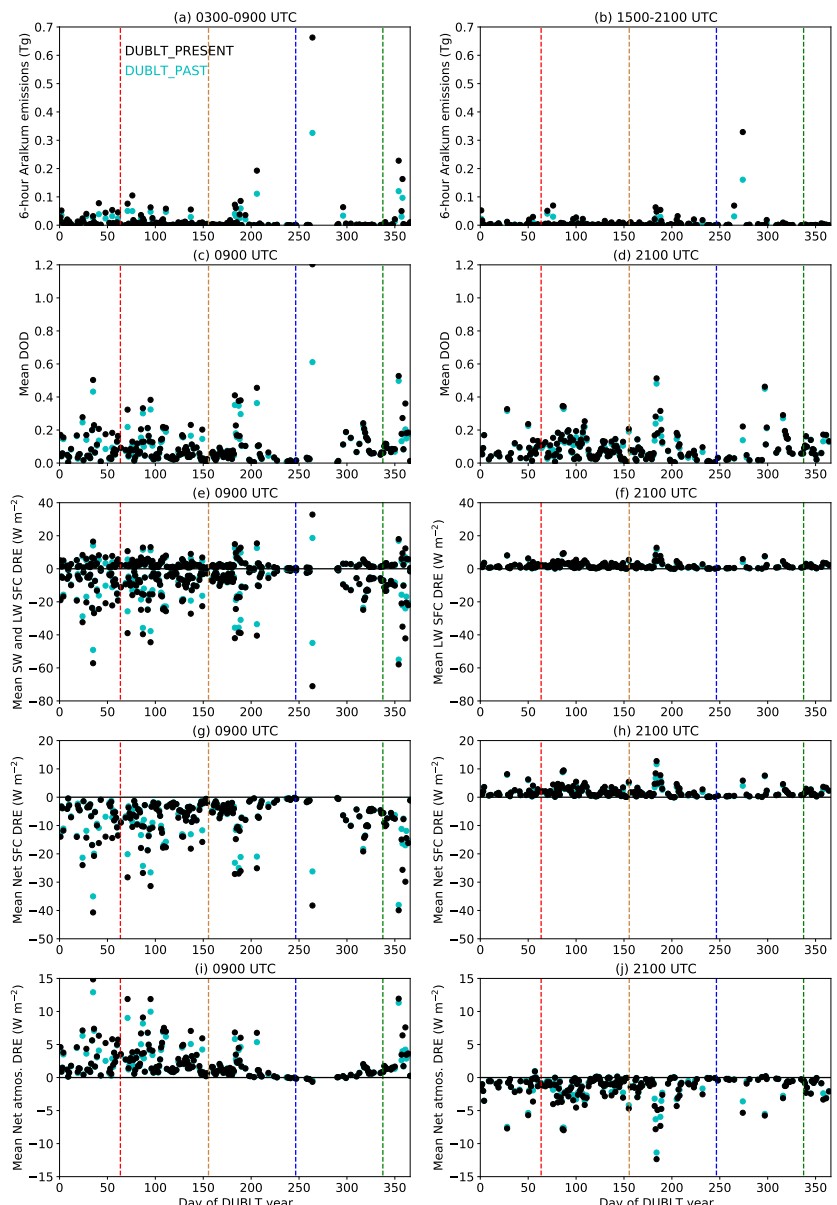

**Figure 4.** Timeseries plots of 6-hourly accumulated dust emissions, instantaneous DODs and dust DREs averaged over the Aralkum for (left) 0900 UTC, and (right) 2100 UTC. 0900 UTC is ~1300 local time, and 2100 UTC is ~0100 local time. Only grid cells with a cloud cover < 1% go into the data subset for each point in the timeseries. Of the 201 grid cells in the Aralkum, more than 10% of the grid cells must fulfil this criterion in order for the point to go into the timeseries. For the emissions, the values are the totals over the Aralkum box for the preceding 6 hours, i.e. 0300-0900 UTC and 1500-2100 UTC. In panel (e) points with negative values are SW, positive values indicate LW. The black points represent the baseline Present DUBLT scenario and the cyan points represent DUBLT_PAST. The dashed vertical lines denote the change of the seasons: the red line is the start of summer (1st June), the brown line the start of autumn, the blue line the start of winter, and the green line the start of the second spring (1st March 2016).

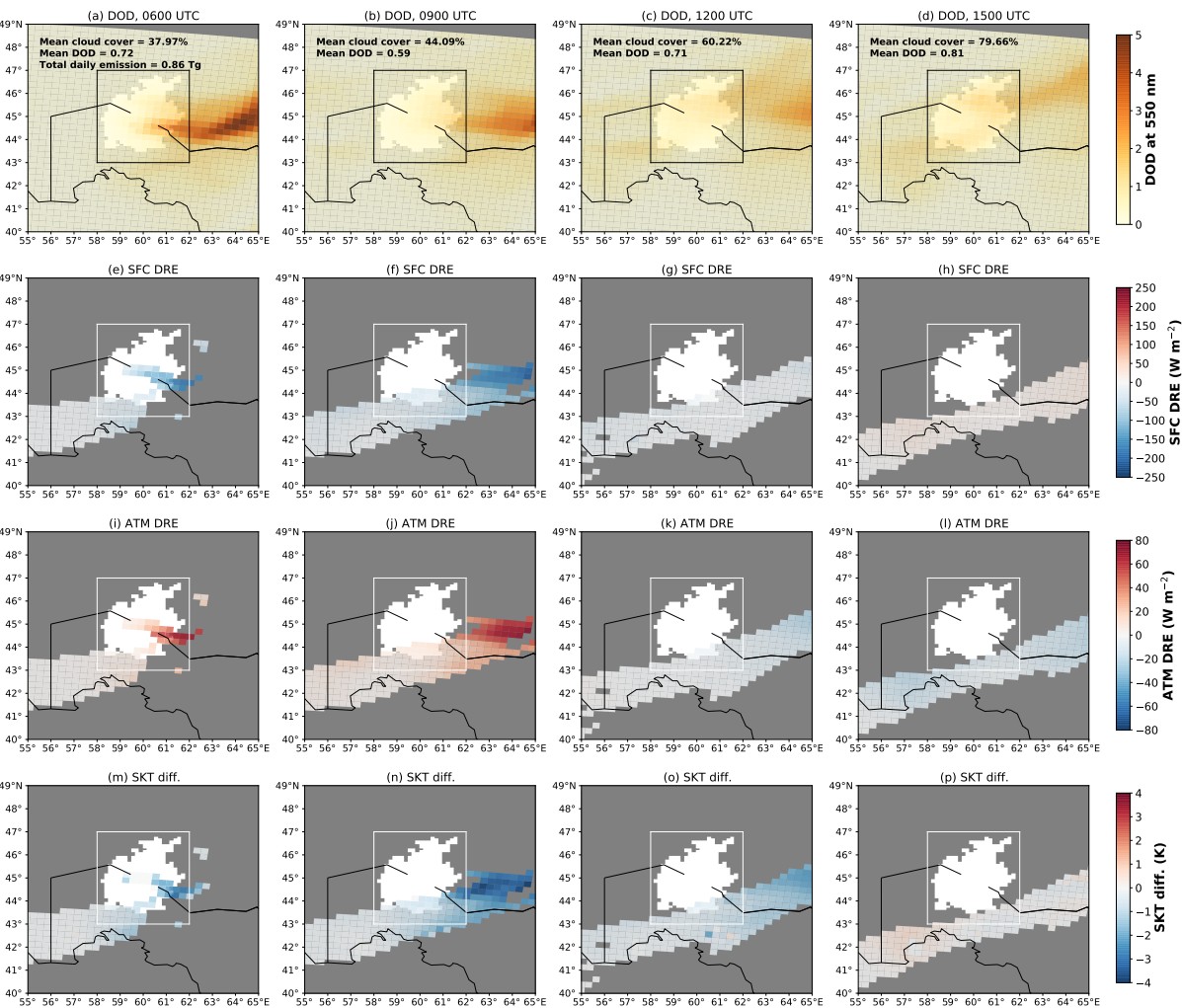

**Figure 5.** Case study of an Aralkum dust event on 17th March 2016, 3-hourly from 0600 to 1500 UTC left-right, modelled by the DUBLT_PRESENT scenario. The first row shows the instantaneous DODs, the second row the SFC DREs, the third row the atmospheric DREs, and the fourth row the differences in the skin temperature (SKT) with respect to the non-dust case. Grey grid cells have a cloud cover $\geq 1\%$, and hence are regarded as missing data. The stated values of the mean cloud cover and DOD, and the total daily emission, are for the entire region of the Aralkum within the square box (43-47°N, 58-62°E).

to the time-of-day as well as in relation to the DODs and cloud cover. Figure 5 depicts the radiative effects of a notable Aralkum dust event from 17th March 2016, a day which contributed to the strong accumulated dust emissions driven by the westerly winds that were characteristic of the spring months of the simulation year (see Figure 10 in BHS22). This day had a total Aralkum dust emission of 0.86 Tg (the day with the sixth highest emissions during the year), and was the third day of a three-day period from 15-17th March which is preeminent amongst Aralkum dust events during the DUBLT simulation year.

This period was responsible for Aralkum dust emissions of 4.99 Tg out of a yearly total of 27.1 Tg, and had the two highest ranked days for Aralkum dust emission during the year. These emissions were, however, co-located with extensive cloud cover except on the third day of the event when the daily average cloud cover over the Aralkum reduced to 69.8%, an opening which permits a look at dust's radiative effects. Four timeslots are mapped in Fig. 5. Dust emissions occurred primarily during the morning hours, driven by westerly winds and transported eastwards over southern Kazakhstan. The earliest and thickest portion of the dust event was co-located with cloud, hindering the radiation simulations, however sufficient dust was still present in the cloud-free areas such that its effects can be discerned.

During 17th March 2016, the maximum surface radiative cooling and atmospheric heating occurred in the early afternoon at 0900 UTC, with a minimum instantaneous SFC DRE of -201 W m$^{-2}$ and a maximum ATM DRE of +68 W m$^{-2}$, resulting in a maximum skin temperature (SKT) cooling of -3.8 K. Meanwhile from left to right the diurnal cycles in the radiative effects are clearly hinted at, with SFC cooling and ATM heating during the day giving way to SFC heating and ATM cooling later into the afternoon and into the night. The ATM DRE appears to flip sign earlier in the afternoon than does the SFC DRE, indicative of the stronger SW SFC effect during the day, compared to both the LW effect and the daytime SW atmospheric absorption. There is more of a lag in the skin temperature response compared to the radiative fluxes, such that some SKT cooling is still apparent in the early evening (Fig. 5(p)) to the south and east of the Aralkum.

The physics of the general situation over the Aralkum region is described in the maps of aggregated yearly mean SFC and ATM DREs in Fig. 6, subdivided and binned by ranges of the solar zenith angle (SZA). The top row indicates the nighttime situation, while the descending rows refer to lower SZA ranges, i.e. higher solar elevation. It is self-evident that under pristine-sky conditions the surface and the atmosphere are heated when the sun is high during the day and cooled during the night: these are quantified in the labelled fluxes, indicating the consequences of the surface properties for the overall radiative environment, and are considered here for reference. At the surface, the nighttime heating effect of dust appears to transition to the daytime cooling effect at an SZA of ~75° (Fig. 6, panels (c) and (e)), with the most substantial SFC cooling effects due to dust occurring over Turkmenistan's Karakum Desert at lower SZAs (Fig. 6(i)). Closely related to this, the Karakum is also where the largest atmospheric heating effects due to dust are simulated (Fig. 6(j)), a consequence of relatively frequent dust activity and Turkmenistan's southern location within the region. Meanwhile the nighttime cooling effect of dust in the atmosphere transitions to the daytime heating effect approximately within the 60-75° range (Fig. 6, panel (f)), due also to the transition in the pristine-sky net flux, with an apparent north-south contrast in cooling and heating over the Kyzylkum and the Karakum, respectively.

## 4  Changes in dust radiative effects due to the expansion of the Aralkum

The extent to which the expansion of the Aralkum has perturbed the regional radiation balance depends on the magnitude of the increase in dust presence and also on the temporal distribution of the additional dust. As reported in BHS22, within the simulations the dust emissions from the Aralkum box increased from 14.3 Tg year$^{-1}$ in the DUBLT_PAST scenario (from the barren land around the Aralkum and that portion of the Aral Sea that had already dried out by the 1980s and 1990s) to

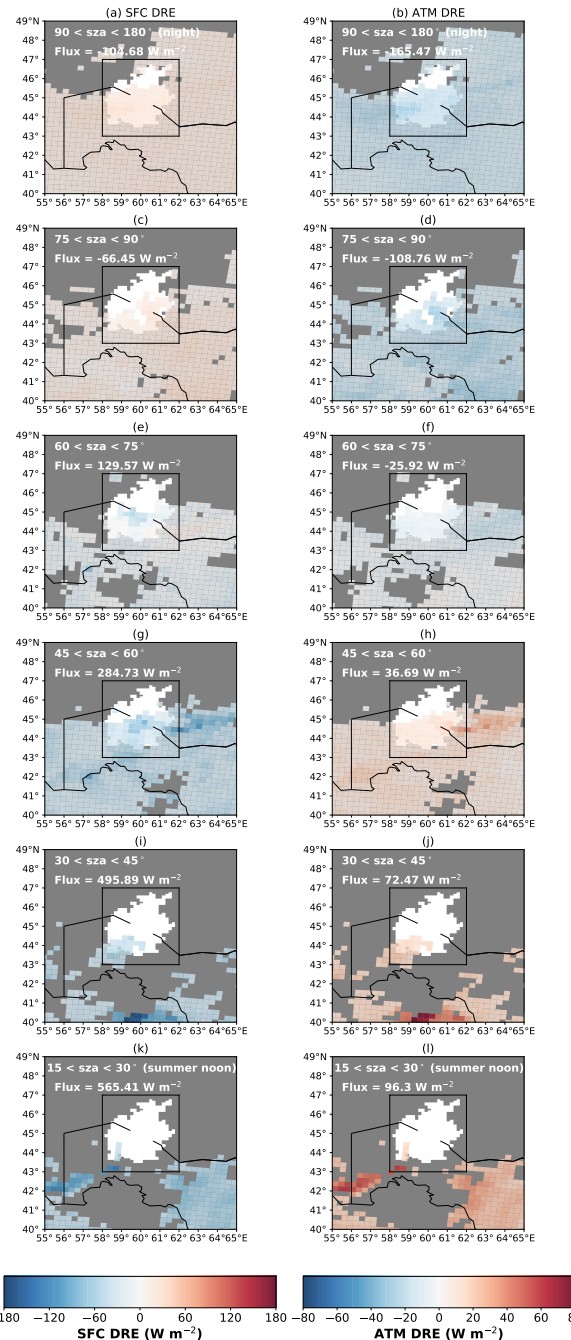

**Figure 6.** Maps of Present dust SFC (left column) and ATM (right column) DREs for DODs $> 0.5$, cloud-screened in both the DUBLT_PRESENT and DUBLT_NODUST scenarios to a maximum cloud cover of 1%, from the entire DUBLT simulation year. Each row is for a specified SZA range, as indicated (one nighttime bin in the top row, 15° SZA bins during the day). The marked fluxes are the pristine-sky mean net fluxes in the DUBLT_NODUST scenario, i.e. excluding dust and cloud.

27.1 Tg year$^{-1}$ in the DUBLT_PRESENT scenario, a doubling of the overall emissions. What this means for the monthly mean DODs over Central Asia is depicted in Fig. 7, which also displays the prominence of the Karakum as a dust source during spring and summer, while the Aralkum becomes the more dominant regional source in December and March. The maps of DUBLT_PRESENT - DUBLT_PAST DOD differences are particularly instructive as to the extra dust that would not be in the atmosphere were it not for the desiccation of the Aral Sea. These differences highlight also November (Fig.

7(q)) as a month with extra dust provided by the Aralkum. In contrast, a reduction in dust presence is apparent especially in October (Fig. 7(p)) over western Turkmenistan and the Caspian Sea, identifying the Garabogazköl Basin as a dust source during the 1980s prior to its re-inundation (see also BHS22 for further details). For simplicity, with the influence of the Garabogazköl Basin as a relevant caveat, the DUBLT_PRESENT - DUBLT_PAST difference will henceforth be referred to as the 'DUBLT_ARALKUM' scenario. The Aralkum dust is not added to a pristine dust-free regional environment, but to a

region including pre-existing dust-producing deserts.

### 4.1 Dust radiative effects in relation to the solar zenith angle and the season

The radiative effects of dust in the atmosphere are dependent on the solar elevation, as previously indicated by Fig. 6. The patterns of the SW and LW DREs over the Aralkum with respect to the associated SZAs and DODs are described in more detail by Fig. 8, both at the surface and in the atmosphere. Depicted here are the DUBLT_PRESENT DREs. It is to be expected

that the largest absolute values of the DREs will be at higher DODs, and during the daytime at lower SZAs, especially in the case of the SW effect. Summarising the contrasting SW and LW effects, in the SW there is no effect at night while during the day at higher DODs dust causes SW SFC cooling and ATM heating. Meanwhile in the LW there is much less dependence on the time-of-day, with dust causing LW ATM cooling and SFC heating, effects which are enhanced by higher DODs. The SW effect clearly dominates the LW effect during the daytime at lower SZA values in spring, summer, and autumn, when the solar

insolation is highest. The dominance of the SW effect during the daytime is especially apparent in spring (see Fig. 8, panels (a) and (c)) at high DODs in the SZA bin between 50 and 60°, with SW SFC bin cooling of -220.8 W m$^{-2}$ and SW ATM bin heating of +81.8 W m$^{-2}$. At night, only the LW effect exists.

Figure 9 expands the analysis of the DREs plotted in Fig. 8 to summarise the DUBLT_PRESENT net DREs alongside the DUBLT_ARALKUM net DRE differences, so as to highlight the impact of the additional Aralkum dust. In general the net

DREs are dominated by the SW effects during the daytime, while the LW effects dominate at night and when the sun is low above the horizon. The maximum bin SFC cooling and ATM heating occurs during the spring, due to the SW effect described in the previous paragraph: there is daytime net SFC cooling in the Present scenario of up to -147.5 W m$^{-2}$ and net ATM heating of +59.6 W m$^{-2}$, both in the 50-60° SZA bin, primarily due to the March 2016 dust events. The equivalent high positive DOD difference bin in the DUBLT_ARALKUM case gives an extra SFC cooling due to Aralkum dust of -67.7 W m$^{-2}$, and an extra

ATM heating of +39.1 W m$^{-2}$. Given the doubling of the Aralkum's emissions from Past to Present, the ratios of these DRE maxima imply that the ATM DREs are slightly more responsive to the presence of Aralkum dust compared to the SFC DREs.

There are daytime bins with higher DODs in winter (during this season the daytime SZAs are higher over the Aralkum) than in the spring, especially when considering the Present with respect to the Past: however these high DOD bins are in the

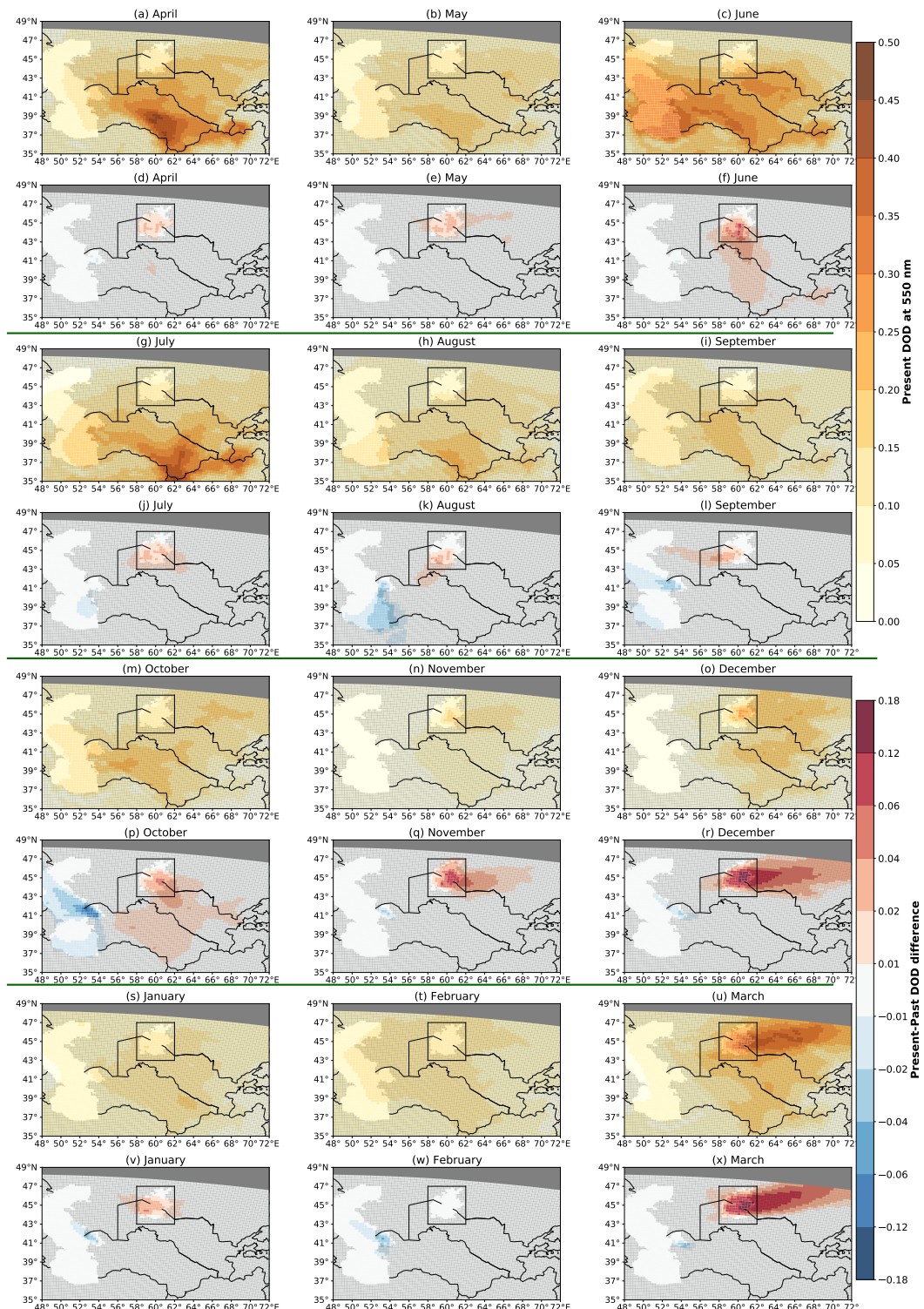

**Figure 7.** Maps of monthly mean DUBLT_PRESENT DODs over Central Asia, along with the associated mean DUBLT_PRESENT - DUBLT_PAST DOD differences (DUBLT_ARALKUM). Panels (a-c), (g-i), (m-o), and (s-u) represent the DUBLT_PRESENT DODs, with the corresponding monthly mean DUBLT_PRESENT - DUBLT_PAST DOD differences plotted in the panels directly beneath.

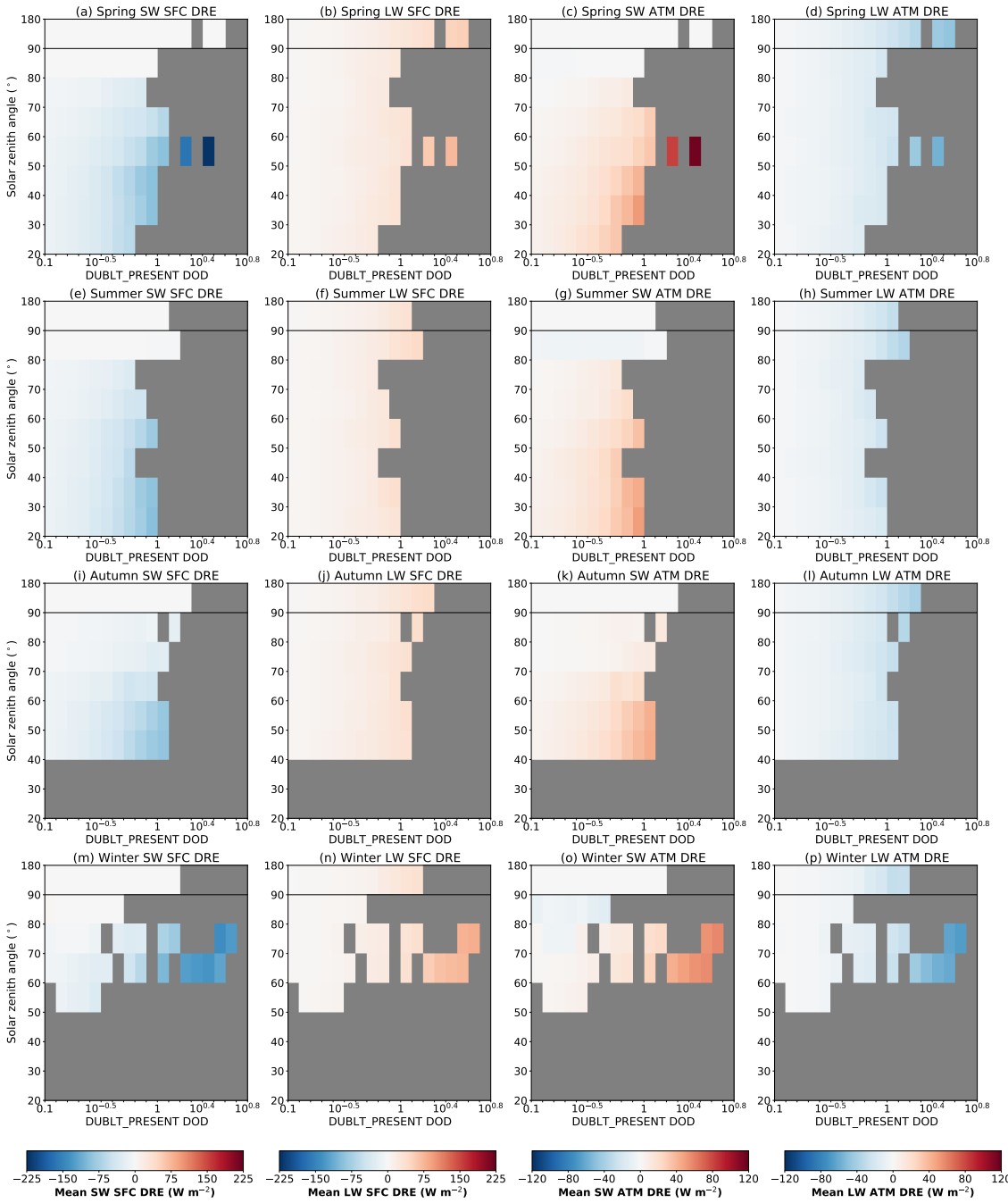

**Figure 8.** Seasonal mean DUBLT_PRESENT SW and LW SFC (left columns) and ATM (right columns) DREs, binned as functions of DUBLT_PRESENT DODs, and of SZA for each grid cell and timeslot within the Aralkum. The Aralkum is here bound by the longitudes 58-62°E and by the latitudes 43-47°N, see also the box in Fig. 1 and subsequent maps. The seasons are MAM, JJA, SON, and DJF. The SZA bin width is 10° during the day, with one bin at night: the solid black horizontal line denotes the day-night boundary. Meanwhile the DOD bin widths are logarithmic, and the SFC and ATM columns have separate colour scales. The allowable cloud threshold is 1% in all three simulations.

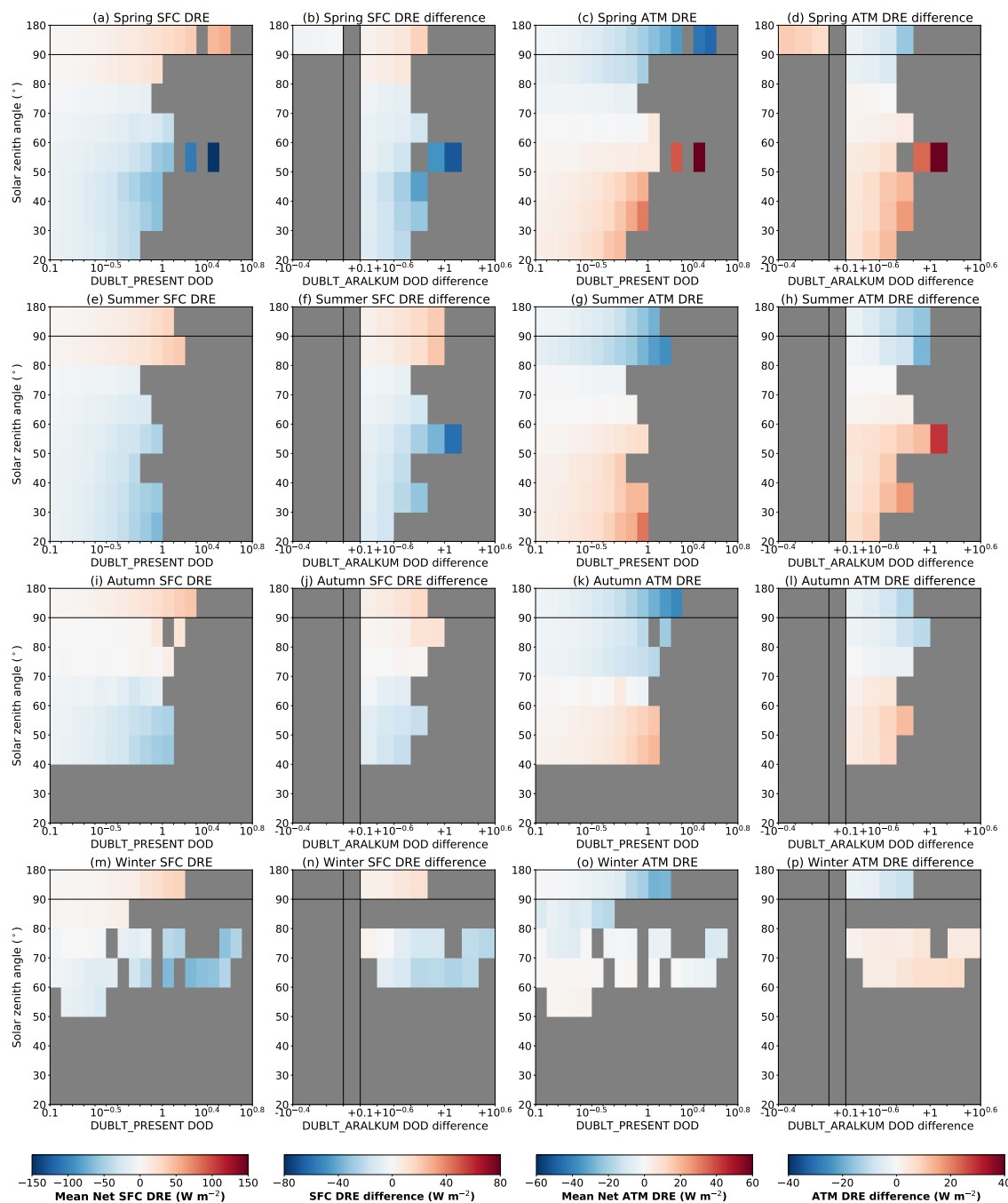

**Figure 9.** Similarly to Fig. 8, seasonal mean DUBLT_PRESENT and DUBLT_PRESENT - DUBLT_PAST (DUBLT_ARALKUM) net SFC (left columns) and ATM (right columns) DREs, binned as functions of DUBLT_PRESENT DODs (first and third columns) and DUBLT_Aralkum DOD differences (second and fourth columns), and of SZA for each grid cell and timeslot within the Aralkum. Each column has its own colour scale.

transition region of the SZA range in which there is more ambiguity as to whether the atmospheric SW heating or LW cooling effect would dominate, see also Fig. 8, panels (o) and (p). These SZA values are where both the surface and atmosphere may cool in the same bins. In the Present scenario the greatest net SFC cooling in winter is -67.8 W m$^{-2}$, but at a DOD of just 1.6-2.0 in the 60-70° SZA bin. Meanwhile the greatest net ATM heating in winter is just +2.4 W m$^{-2}$ within the DOD bin from 0.25-0.32 (understood logarithmically as the range from $10^{-0.6}$-$10^{-0.5}$) and an SZA from 50-60°. The highest daytime DOD bins ($> 2.0$, logarithmically $> 10^{+0.3}$) in winter have net DREs which are cooling both at the surface and in the atmosphere. More dust does not automatically imply a greater radiative effect.

At night, the Present DREs have opposite signs compared to during the day, due to the controlling LW effect, i.e. SFC warming and ATM cooling. This is also shown by the second and fourth columns of Fig. 8. For the nighttime spring is again the season with the highest Present DODs and DREs, although it appears also to be the season with the relatively uncommon occurrence of higher Past DODs compared to the Present. This is due to other formerly dry lakebeds such as the Garabogazköl Basin. In the Present scenario the maximum bin SFC nighttime warming is +53.9 W m$^{-2}$ (compared to minimum SFC daytime cooling of -147.5 W m$^{-2}$), while the minimum ATM cooling is -47.2 W m$^{-2}$ (compared to maximum daytime ATM warming of +59.6 W m$^{-2}$). This quantifies the relatively weak LW effect at night compared to the stronger and competing SW and LW effects during the day. Meanwhile the additional Aralkum dust (Present-Past) adds up to an extra +20.5 W m$^{-2}$ of SFC heating in the summer, and an extra -15.7 W m$^{-2}$ of ATM cooling also in summer. The additional effects of Aralkum dust appear to be less significant at night than during the daytime.

Analysing this information from a different perspective, Fig. 10 investigates further the significance of the impacts of Aralkum dust on the overall radiation budget. Figure 10 seeks to elucidate the changes in the distributions of the radiative effects between the Past and Present by considering their distributions on the monthly timescale, along with the distributions of the DODs and the Present DREs weighted by the DOD (the 'DRE efficiency'). By considering the DRE efficiencies it is thereby possible to extract the seasonal cycle of the DODs from that of the DREs. The plots are subdivided into nighttime and daytime data subsets, emphasising the distinctions between the cooling and warming modes and the primacy of either the SW or LW effects: as shown by Figs. 6 and 9, the net dust radiative effects are negligible when the sun is low above the horizon. June and October appear here to be particularly dusty months in general over the Aralkum in the Present scenario (Fig. 10(a)), due to Aralkum dust (Fig. 10(b)), but December and March are the months with the biggest individual events. In most months dust cools the surface during the day and warms it at night, and warms the atmosphere during the day and cools it at night; Aralkum dust (denoted by the Present-Past differences) typically acts to enhance these patterns. In general, the strongest persistent Present and Present-Past DREs occur during the summer and spring months when there is the most solar and thermal radiation available for dust to interact with, a point which is emphasised by the DRE efficiencies in Fig. 10, panels (g) and (h). At the surface dust is more efficient in its SW cooling effect than it is in its LW heating effect. It is also clear that there is a stronger seasonal cycle in the DRE efficiencies during the daytime compared to during the night, as a consequence of the seasonal variations in the solar zenith angle.

Diverting briefly to consider the totality of this particular year, Tables 1 and 2 show that at the surface the daytime radiative cooling effect dominates the overall effect. Meanwhile in the atmosphere the nighttime cooling effect is also dominant com-

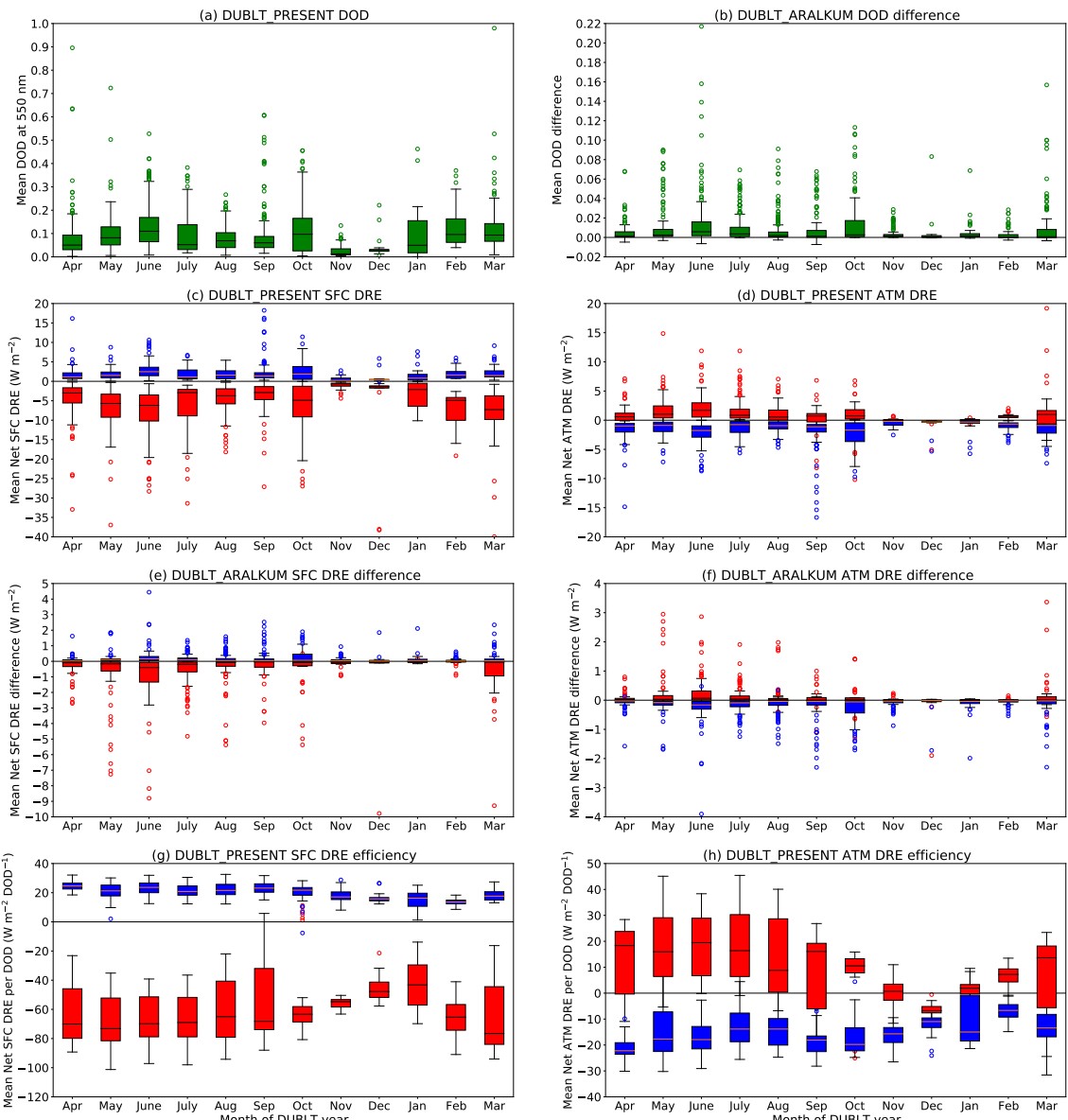

**Figure 10.** Boxplots displaying (top row) the monthly DOD values averaged over the cloud-free grid cells of the Aralkum (per grid cell the maximum allowable cloud threshold is 1% in both the dust and the NODUST scenario, with $> 10\%$ of grid cells required to be cloud-free for a timeslot to be included in the monthly statistics), in the Present scenario, and their Present-Past (DUBLT_ARALKUM) differences, with outliers included for points beyond the quartile ranges. In the second and third rows are the associated monthly net (left column) SFC and (right column) ATM DRE values. In the bottom row are the corresponding ranges for the DREs weighted by the DODs. For the DREs the timeslots are subdivided by time-of-day: the blue bars and outliers (marked by circles) denote the nighttime timeslots, with an SZA $\geq 90°$, while the red bars and outliers denote the daytime timeslots, here defined as having an SZA $< 75°$. Therefore daytime timeslots at low solar elevation are excluded. The y-axes are truncated for clarity, but there are outliers. In panel (a) there are outliers in December at DODs of 1.20 and 1.78, while in panel (b) there are outliers in December at +0.59 and +0.92, and in March at +0.42. In panel (c) there are outliers in April at +23.8 W m$^{-2}$ (night), in May at -40.7 W m$^{-2}$ (day), and in March at -61.8 W m$^{-2}$ (day). In panel (d) there is a nighttime outlier in April at -21.1 W m$^{-2}$. In panel (e) there are daytime outliers in December at -12.0 W m$^{-2}$ and in March at -12.9 and -23.0 W m$^{-2}$. In panel (f) there is a daytime outlier in March at +8.9 W m$^{-2}$.

**Table 1.** Accompanying Fig. 10, the yearly mean Aralkum area-averaged and cloud-screened values of the DODs and the SFC, ATM and TOA DREs (net, SW, and LW), for both the DUBLT_PRESENT and DUBLT_ARALKUM scenarios, along with their associated standard deviations. The data are first averaged spatially over the Aralkum, the means and the standard deviations are then calculated over the $n_{days} \times n_{hours}$ timeslots during the year. 'Day' is at an SZA $< 75°$ and night is at an SZA $\geq 90°$. DREs have units $W\,m^{-2}$.

| | All | Day | Night |
|---|---|---|---|
| (a) DUBLT_PRESENT DOD | 0.099±0.107 | 0.103±0.126 | 0.094±0.091 |
| (b) DUBLT_PAST DOD | 0.089±0.085 | 0.090±0.090 | 0.087±0.084 |
| (c) DUBLT_ARALKUM DOD | 0.0097±0.0339 | 0.0130±0.0489 | 0.0068±0.0134 |
| (d) DUBLT_PRESENT net SFC DRE | -1.34±6.19 | -6.09±6.64 | +2.03±2.33 |
| (e) DUBLT_PRESENT SW SFC DRE | -3.90±7.95 | -9.26±10.03 | 0.00±0.01 |
| (f) DUBLT_PRESENT LW SFC DRE | +2.56±3.09 | +3.17±3.77 | +2.03±2.33 |
| (g) DUBLT_ARALKUM net SFC DRE | -0.15±1.19 | -0.64±1.64 | +0.20±0.42 |
| (h) DUBLT_ARALKUM SW SFC DRE | -0.46±1.86 | -1.08±2.76 | 0.00±0.00 |
| (i) DUBLT_ARALKUM LW SFC DRE | +0.31±0.88 | +0.44±1.22 | +0.20±0.42 |
| (j) DUBLT_PRESENT net ATM DRE | -0.62±2.91 | +1.24±2.25 | -1.55±2.12 |
| (k) DUBLT_PRESENT SW ATM DRE | +1.13±3.37 | +3.21±4.14 | 0.00±0.03 |
| (l) DUBLT_PRESENT LW ATM DRE | -1.75±2.45 | -1.97±2.83 | -1.56±2.12 |
| (m) DUBLT_ARALKUM net ATM DRE | -0.05±0.51 | +0.15±0.54 | -0.19±0.38 |
| (n) DUBLT_ARALKUM SW ATM DRE | +0.20±0.87 | +0.48±1.29 | 0.00±0.00 |
| (o) DUBLT_ARALKUM LW ATM DRE | -0.26±0.71 | -0.34±0.97 | -0.19±0.38 |
| (p) DUBLT_PRESENT net TOA DRE | -1.96±4.19 | -4.85±5.09 | +0.47±0.45 |
| (q) DUBLT_PRESENT SW TOA DRE | -2.77±4.91 | -6.05±6.10 | 0.00±0.03 |
| (r) DUBLT_PRESENT LW TOA DRE | +0.81±0.88 | +1.20±1.09 | +0.47±0.45 |
| (s) DUBLT_ARALKUM net TOA DRE | -0.20±0.83 | -0.50±1.23 | +0.01±0.06 |
| (t) DUBLT_ARALKUM SW TOA DRE | -0.26±1.01 | -0.60±1.49 | 0.00±0.00 |
| (u) DUBLT_ARALKUM LW TOA DRE | +0.06±0.19 | +0.10±0.27 | +0.01±0.06 |

pared to the daytime heating, with Aralkum dust enhancing both of these effects. There are slightly higher DODs during the day
than during the night in the Present scenario (Table 1(a)), while dust from the Aralkum (c) is added particularly prominently to
the (cloud-free) atmosphere over the Aralkum during the daytime compared to during the night. In the Past scenario (b) there
were much more balanced average dust loadings between day and night. This relates also to the apparent behaviour of evening
dust emissions occurring more frequently under cloudy skies which was previously noted in relation to Fig. 4: such dust load-
ings would be more frequently cloud-screened for the purposes of analysing the dust DREs. In the Present scenario daytime

cooling at the surface (d) is comparatively stronger than daytime heating in the atmosphere (j), while the absolute values of the nighttime SFC and ATM effects are broadly equivalent with each other. Aralkum dust adds an extra -0.15±1.19 W m$^{-2}$ (standard deviation denoted by ±) of cooling at the surface (g), up to -23.0 W m$^{-2}$ on an event basis. Aralkum dust also adds an extra -0.05±0.51 W m$^{-2}$ of cooling in the atmosphere (m), with extrema of -5.82 and +8.91 W m$^{-2}$. The competing SW heating and LW cooling effects on the atmosphere result in a net cooling of the atmosphere to a value of -0.62±2.9 W m$^{-2}$ on

the yearly timescale (j), with daytime SW heating of the atmosphere of +3.21 W m$^{-2}$ (k) being outweighed by the cumulative daytime and nighttime LW cooling (l). During the daytime there is a net heating of the atmosphere by +1.24 W m$^{-2}$ due to the greater intensity of the SW effect compared to that of the LW.

**Table 2.** Accompanying Table 1, the minimum and maximum Aralkum area-averaged and cloud-screened values of the DODs and the SFC and ATM DREs (DUBLT_PRESENT and DUBLT_ARALKUM). DREs have units W m$^{-2}$.

|  | Minimum | Maximum |
|---|---|---|
| DUBLT_PRESENT DOD | 0.00 | 1.78 |
| DUBLT_PAST DOD | 0.00 | 0.86 |
| DUBLT_ARALKUM DOD | -0.01 | +0.92 |
| DUBLT_PRESENT SFC DRE | -61.8 | +23.8 |
| DUBLT_ARALKUM SFC DRE | -23.0 | +5.98 |
| DUBLT_PRESENT ATM DRE | -21.1 | +19.2 |
| DUBLT_ARALKUM ATM DRE | -5.82 | +8.91 |

It is important to note here the likelihood of lagged effects due to the dust interaction with radiation, which may be obscured by the long-term averages of the instantaneous radiative effects considered here. SW SFC cooling during the daytime

suppresses the ground temperatures (see Fig. 5(m-p)) and would thereby reduce the upwelling LW radiation from the surface over subsequent hours, on occasion lasting also into the night and subsequent days. This would trigger atmospheric temperature adjustments in response (e.g. Miller, 2012; Stjern et al., 2023), possibly also including cloud adjustments, which would perturb the atmospheric radiation balance from the non-dust case even in the absence of further dust outbreaks. This process may contribute to apparent instantaneous dust heating of the surface and cooling of the atmosphere. The influence of the dust

impacts on the atmospheric environment will be discussed further in Section 5.

Returning to the seasonal cycle, it is again clear that in the Present scenario the daytime SFC cooling effect (Fig. 10(c)) is more significant than both the nighttime SFC heating effect and the ATM (Fig. 10(d)) cooling and heating effects. Similarly the Aralkum's dust contributes most to enhanced SFC daytime cooling (Fig. 10(e)), especially in June and March (see also Table 3). Meanwhile the Aralkum's daytime and nighttime atmospheric effects (Fig. 10(f)) are fairly well balanced. During winter the

DUBLT_PRESENT and DUBLT_ARALKUM DREs are comparatively weak, especially in November and December when only very occasional dust outbreaks perturb the regional radiation environment. December in particular is an unusual month

**Table 3.** Accompanying Table 1, the monthly mean Aralkum area-averaged and cloud-screened values of the DODs and the SFC and ATM DREs (DUBLT_PRESENT and DUBLT_ARALKUM), along with their associated standard deviations. DREs have units $W\,m^{-2}$.

| Month | DUBLT_PRESENT DOD | SFC DRE | ATM DRE | DUBLT_ARALKUM DOD | SFC DRE | ATM DRE |
|---|---|---|---|---|---|---|
| April | 0.090±0.123 | -1.71±6.35 | -0.47±3.05 | 0.005±0.011 | -0.15±0.55 | -0.01±0.22 |
| May | 0.102±0.088 | -2.85±7.18 | +0.12±2.89 | 0.011±0.021 | -0.36±1.42 | +0.03±0.60 |
| June | 0.131±0.097 | -1.98±7.78 | -0.76±4.08 | 0.015±0.027 | -0.26±1.46 | -0.08±0.76 |
| July | 0.092±0.080 | -1.64±6.14 | -0.36±3.07 | 0.009±0.013 | -0.15±0.74 | -0.04±0.34 |
| August | 0.078±0.052 | -1.31±4.27 | -0.17±1.87 | 0.007±0.013 | -0.11±0.78 | -0.04±0.35 |
| September | 0.090±0.103 | -0.32±4.69 | -1.05±2.88 | 0.007±0.014 | -0.01±0.66 | -0.10±0.39 |
| October | 0.123±0.118 | -0.45±6.46 | -1.49±3.01 | 0.014±0.024 | +0.02±0.97 | -0.19±0.54 |
| November | 0.026±0.026 | +0.19±1.26 | -0.43±0.81 | 0.004±0.007 | +0.04±0.27 | -0.07±0.17 |
| December | 0.094±0.299 | -1.40±7.77 | -0.61±1.09 | 0.034±0.154 | -0.41±2.22 | -0.10±0.36 |
| January | 0.099±0.105 | +0.63±3.38 | -1.42±3.11 | 0.005±0.011 | +0.11±0.34 | -0.11±0.32 |
| February | 0.123±0.072 | -0.51±4.75 | -1.20±2.65 | 0.003±0.005 | +0.02±0.17 | -0.04±0.10 |
| March | 0.124±0.113 | -2.59±8.72 | -0.56±3.01 | 0.013±0.044 | -0.47±2.60 | +0.04±0.96 |
| Year | 0.099±0.107 | -1.34±6.19 | -0.62±2.91 | 0.0097±0.0339 | -0.15±1.19 | -0.05±0.51 |

for the Aralkum, as being the one month in the year when dust typically cools the atmosphere both during the night and during the day (although there are also daytime cooling outliers in other months especially in September, October, and March). Correspondingly, in December during the day dust acts to cool both the surface and the atmosphere. The unusual nature of the December effect is particularly prominent in the DRE efficiencies (Fig. 10(h)).

## 4.2 The influence of dust type on the overall radiative effects

The outcome that dust over the Aralkum has a simulated net cooling effect on the atmosphere is a consequence of the relatively reflective optical properties of the modelled dust type, within a limited-area mid-latitude region. This contrasts with previous estimates of dust radiative effects on the atmosphere (e.g. Miller et al., 2014) which indicate a typical net heating of the atmosphere due to dust, on larger regional to global scales. How appropriate then is the choice of dust type within the DUBLT modelling system? Measurements of dust aerosol chemical composition made by Fomba et al. (2019) at Dushanbe in Tajikistan led to the finding that the calcium-iron ratio of dust is twice as high in Dushanbe compared to similar measurements from the Sahara. It can be inferred from this that Central Asian dust is significantly saltier than Saharan dust, and therefore that Central Asian dust would be more optically scattering (see also Hofer et al. (2020) and Xi (2023)). Hence it is reasonable to choose a more optically scattering dust type for Central Asia, compared to a more absorbing dust type, both of which could be simulated by COSMO-MUSCAT (Helmert et al., 2007).

An estimate of the DUBLT modelling system's uncertainty on the radiative effects of Aralkum dust can be inferred from the differences between the scattering and absorbing scenarios, as shown in Table 4. Broadly the same surface cooling pattern exists in both scenarios, with a stronger cooling effect triggered by more absorbing dust. Meanwhile in the atmosphere there is

a difference in sign between the scenarios: as stated before the scattering scenario leads to a net cooling of -0.62±2.91 $\mathrm{W\,m^{-2}}$, while the absorbing scenario results in a net heating of +0.96±5.48 $\mathrm{W\,m^{-2}}$. The difference between the scenarios in the atmosphere is explained both by the intensity of the daytime heating, and by the duration of the daytime mode. With reference to the third column of Fig. 9 it was found that the threshold solar zenith angle (between 'daytime' and 'nighttime' modes) for the scattering dust was ~70°, while the equivalent threshold for the absorbing dust (not plotted) is higher at ~80°. There

is therefore a longer daytime duration of atmospheric heating in the absorbing scenario, with the overall outcome that the scattering and absorbing scenarios have opposite signs on the yearly timescale.

**Table 4.** The yearly mean Aralkum area-averaged and cloud-screened values of the DUBLT_PRESENT and DUBLT_ABS TOA, ATM and SFC DREs ($\mathrm{W\,m^{-2}}$) along with their associated standard deviations, in comparison with values from other studies. The DRE is also known as the radiative forcing. The values from the DUBLT simulations and from Xie et al. (2018) are the clear-sky values, while for Albani et al. (2014) and Scanza et al. (2015) only all-sky values (i.e. including cloud) are available.

|  | TOA DRE | ATM DRE | SFC DRE |
| --- | --- | --- | --- |
| DUBLT_PRESENT |  |  |  |
| Scattering | -1.96±4.19 | -0.62±2.91 | -1.34±6.19 |
| Absorbing | -0.98±2.73 | +0.96±5.48 | -1.95±7.85 |
| Albani et al. (2014), Table 7 |  |  |  |
| C4fn | -0.23±0.02 | +0.33±0.01 | -0.56±0.03 |
| Scanza et al. (2015), Table 8a |  |  |  |
| CAM4-m | +0.05 | +0.23 | -0.18 |
| CAM5-m | +0.05 | +0.67 | -0.62 |
| Xie et al. (2018), Table 5 |  |  |  |
| East Asia | -1.71 | +2.88 | -4.59 |
| North Africa | +5.11 | +3.45 | +1.66 |

Table 4 also presents comparisons of the DUBLT DREs over the Aralkum with results from previous studies (Albani et al., 2014; Scanza et al., 2015; Xie et al., 2018) over other regions of the globe. Dust radiative effects over Central Asia were also simulated by Li and Sokolik (2018), using the WRF-Chem-DuMo model, however these were performed on a one-

day dust event case study in May 2007 and so are not comparable with the yearly timescale. It is clear that the scattering dust from the Aralkum is an outlier in terms of its atmospheric cooling effect, at -0.62 $\mathrm{W\,m^{-2}}$ compared to +0.33 $\mathrm{W\,m^{-2}}$ of atmospheric radiative warming simulated by Albani et al. (2014), +0.23 and +0.67 $\mathrm{W\,m^{-2}}$ by Scanza et al. (2015), and +2.88 and +3.45 $\mathrm{W\,m^{-2}}$ by Xie et al. (2018). The values from these three papers were from global simulations using the Community

Atmosphere Model (CAM-4, and CAM-5) over periods of ten, six, and twenty years, respectively, although for Xie et al. (2018) the radiative forcing values are quoted for the spring (MAM) season only. Each study investigated the sensitivity of the radiation budget to dust: for Albani et al. (2014) it was the difference between the present day and the Last Glacial Maximum (∼21,000 years ago); for Scanza et al. (2015) it was the sensitivity to the dust mineralogy; and for Xie et al. (2018) it was the regional differences between North Africa and East Asia. The higher values published by Xie et al. (2018) may be a consequence of the seasonal timeframe and the regional focus over major dust source regions. It is also apparent that the DUBLT absorbing dust run provides an atmospheric warming of +0.96 W m$^{-2}$ that is within the range of the CAM-4 and CAM-5 simulation values, as an indicator of where the DUBLT modelling system agrees with previous studies. However globally we may expect that dust is typically more absorbing than dust over Central Asia, a region with a particularly high density of desiccated lakes such as the Aralkum: these include Lake Urmia in western Iran (Sotoudeheian et al., 2016) and the Sistan Basin on the border between Afghanistan and Iran (Alizadeh-Choubari et al., 2014).

## 5   Perturbations to the atmosphere due to the presence of Aralkum dust

Given these quantified simulated dust DREs, and the differences between the Past and the Present epochs, what are the implications for the atmospheric state in response to the presence of Aralkum dust aerosol? COSMO-MUSCAT simulates radiative feedback of the dust on the atmospheric state, and so it is therefore possible to make quantitative estimates of the differences in the atmospheric variables between the various scenarios. Figure 11 maps the differences in the skin temperature (SKT), the air temperature at 950 hPa (T950hPa, i.e. ∼500 m above sea level), the pressure at mean sea level (PMSL), and the total cloud cover between the Present and the Past scenarios. This difference between scenarios isolates the effect on the atmosphere of the additional Aralkum dust. It is important to consider these changes primarily on longer timescales given the cumulative perturbations by Aralkum dust to the Central Asian atmosphere, and the associated lag times in the response of the atmosphere to additional dust aerosol. Given the noise in the data, in order to identify signals in the atmospheric variables more clearly Gaussian smoothing has been applied to the data (e.g. Miinalainen et al., 2021, see supporting information), with a $\sigma$ value of 2. These plots should also be considered with reference to the DUBLT_PRESENT DODs and DUBLT_ARALKUM DOD differences mapped in Fig. 7. It is also important to recognise that these mapped perturbations are monthly averages, and so the magnitude of the perturbations may appear modest. Furthermore, it is relevant to note that the perturbations are often larger over the mountain regions to the south-east (in Tajikistan, Afghanistan and Kyrgyzstan) where the orography can introduce supplementary effects: these effects will not be considered in this discussion.

The atmospheric perturbations differ according to the atmospheric dust distribution and the season of the year, and are highly non-linear in their responses, with effects propagating to areas remote from the Aralkum. This non-linearity also reflects the time delay in the responses to dust activity (as shown for other aerosol species by Stjern et al. (2023)). Dust directions from the Aralkum for this simulation year are typically southwards in June (a direction previously noted by other authors including Wang et al. (2022)), westwards in September, and eastwards in December and March. Bearing that in mind, the SKT response to the prevailing dust directions appears to be warming in June, September and December (up to +0.020 K), but cooling in March,

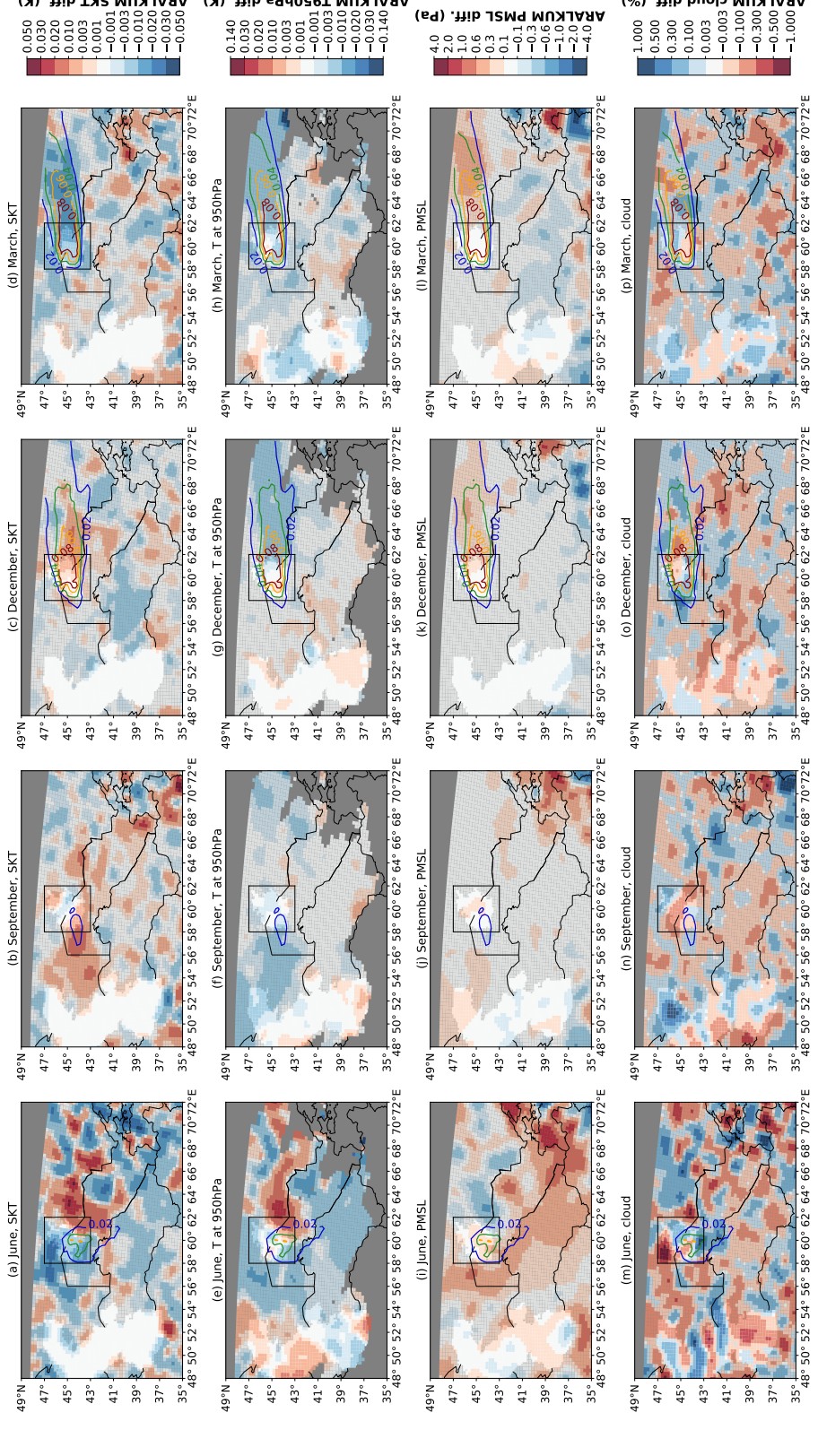

**Figure 11.** Maps of smoothed perturbations (DUBLT_PRESENT - DUBLT_PAST, i.e. DUBLT_PRESENT - DUBLT_PAST, i.e. DUBLT_ARALKUM) of SKT (top row), T950hPa, PMSL, and total cloud cover (bottom row), for June, September, December, and March. Contour lines indicate the corresponding monthly mean DUBLT_ARALKUM DOD values, so as to isolate the spatial distribution of additional Aralkum dust. Gaussian smoothing has been applied to the data ($\sigma$ = 2), note also that the colour scales are non-linear.

down to -0.021 K. Meanwhile the T950hPa response is ambiguous in the hotter month of June, but notably cooling in the cooler months of September, December and March, down to -0.009 K. More consistent patterns seem to be apparent in the PMSL response, typically with increases in the pressure downwind of the Aralkum of up to +0.49 Pa in June, December and March, with the increased pressure pattern being particularly prominent in March. The PMSL response in September is comparatively negligible. The June PMSL pattern (Fig. 11(i)) shows a degree of complexity, despite the increased pressure downwind of the Aralkum, with perturbations nearby to the east of the Aralkum. Positive peaks lie immediately to the west and to the east in northern Uzbekistan (up to +0.76 Pa), but are balanced by a negative trough to the east in southern Kazakhstan (down to -0.64 Pa). Given the prevailing meteorology of the region depicted in Fig. 1, this would therefore imply a strengthening of the Siberian High in the winter and early spring (of the simulation year), and a weakening of the Central Asian Heat Low in summer. Both weather patterns mainly influence the locally prevailing weather conditions and air mass transport and, therefore, also the occurrence of dust storms in Central Asia, but also play an important role in the global atmospheric circulation context. The heat low ventilates the heated summertime land masses of Central Asia by attracting cooler and moist air from the Central Indian Ocean. The Siberian High is the largest high-pressure system that prevails in the northern hemisphere during the winter. An intensification leads to increased pressure gradients and thus to stronger dust-producing surface winds over the Central Asian desert regions.

Pressure and temperature perturbations to the atmospheric environment would also influence cloud formation and lifetime, hence it is also clear that Aralkum dust may have semi-direct effects on the atmosphere (e.g. Meier et al., 2012). It is well established that mineral dust in the atmosphere has a significant role as ice-nucleating particles, INP (e.g. DeMott et al., 2010; Tan et al., 2016): additional dust from the Aralkum may therefore prompt the development of ice and mixed-phase clouds, which would be a further perturbation to the regional climate. While the Aralkum's semi-direct effects on the clouds are predominantly stochastically distributed in nature, a direct correlation with the dust immediately above the Aralkum is apparent. This is particularly the case in June, with a decrease in cloud cover (Fig. 11(m)) in the vicinity of the Aralkum by as much as -0.72%. It can also be seen that the changed cloud cover has a similar magnitude of effect on the surface temperature as the direct radiative influence of the dust. In particular, regions with a significant increase in cloud cover (June and March) are clearly recognisable by a decrease in temperature. Finally, there are also displacements in the precipitation spatial distribution (not shown), with negative perturbations downwind of the Aralkum up to -1.5 mm month$^{-1}$ and positive perturbations up to +1.1 mm month$^{-1}$.

## 6   Conclusions

Building on the findings of BHS22, which described the impact of the desiccation of the Aral Sea in Central Asia on the quantity and distribution patterns of dust emitted from the Aralkum, this current paper investigates the impact of this increase in dust activity on the radiation environment in the vicinity of the Aralkum. Referring back to the questions posed at the end of Section 1, what are the consequences of the expanded Aralkum for the radiative effects of dust in the region? Given that dust activity primarily occurs in the form of discrete dust events, it is clear that on the yearly timescale the perturbation of

the surface and atmospheric radiation environment due to dust may be modest: in the baseline Present scenario the yearly average cloud-screened net SFC dust DRE over the Aralkum is -1.34 W m$^{-2}$ with a standard deviation ($\pm$) of 6.19 W m$^{-2}$, while the average net ATM DRE is -0.62$\pm$2.91 W m$^{-2}$. However during individual dust events the DREs can be markedly more significant, with grid cells experiencing up to -207 W m$^{-2}$ of instantaneous net SFC radiative cooling and +95 W m$^{-2}$ of net atmospheric radiative heating due to dust. Of these values, -116 W m$^{-2}$ of surface cooling and +54 W m$^{-2}$ of atmospheric

heating are due to Aralkum dust.

     What are the patterns of the radiative cooling and warming effects of dust on the surface and atmospheric environment of the Aralkum region? As with other desert regions, the intensity and the sign of the dust DRE over the Aralkum is highly dependent on the time of day and the season during the year in which the dust events occur, primarily as a function of the solar insolation (considered here with reference to the SZA). At the surface of the Aralkum, dust tends to cause radiative cooling during

the daytime and heating during the night, transitioning between the two regimes in an SZA range of $\sim$70-80°. Dust events occurring during these transition periods of the day will have a negligible impact on the surface radiation environment, while the strongest impacts occur at local noon. It is also apparent that dust events lasting between the day and the night (e.g. Fig. 5) will cause opposite radiative impacts at different times during its life-cycle. Meanwhile in the atmosphere dust tends to cause radiative heating during the day and radiative cooling at night, transitioning in the SZA range between $\sim$60-70°. The radiative

effects typically have stronger intensities during daylight (i.e. heating in the atmosphere), however since the transition point is $<$70° the larger fraction of the diurnal cycle is contained within the 'night-time' mode, and hence atmospheric cooling is more frequent to the extent that the overall effect on the atmosphere on the yearly timescale is also cooling (as at the surface). This overall atmospheric cooling effect is a consequence of the optically scattering properties of the assumed dust model, chosen due to the salt content of Central Asian dust.

How have the patterns of the dust DREs changed as the Aralkum has expanded? Considering the changes between Past and Present epochs, the near-doubling in dust emissions over the Aral Sea / Aralkum region (due to the expansion of the barren lakebed) has resulted in increases in the occurrence of both radiative cooling and radiative warming events at the surface and in the atmosphere. These 'new' dust events, that would not have occurred without the desiccation of the Aral Sea, do not occur year-round: instead they occur as episodes during June, September, November, December (weak atmospheric

radiative cooling), and March (occasionally strong atmospheric warming). Compared to the Past scenario ('pre-desiccation'), Aralkum dust is added to the cloud-free atmosphere more heavily during daytime than during nighttime, leading typically to an enhanced cooling effect at the surface (-0.15$\pm$1.19 W m$^{-2}$), clearly outweighing the enhanced heating at night. However in the atmosphere, since the daytime heating effect is weaker than the nighttime cooling effect, additional Aralkum dust also tends to cool the atmosphere on the yearly timescale, by -0.05$\pm$0.51 W m$^{-2}$. Instantaneously, however, there are greater atmospheric

heating than cooling events (up to +8.91 W m$^{-2}$ compared to -5.82 W m$^{-2}$).

     As to whether these radiative effects of Aralkum dust have had consequences for the wider atmospheric environment, these appear to be modest but potentially of some significance. Through the radiative feedbacks contained within the regional modelling approach, it is possible to make quantitative estimates of the perturbations to the atmospheric state due to the addition of Aralkum dust. On the monthly timescale Aralkum dust typically triggers a positive perturbation to the surface pressure

(PMSL) by up to +0.76 Pa in the vicinity of the Aralkum, implying a strengthening of the Siberian High in the colder months and a weakening of the Central Asian Heat Low in summer. The consequent effects on the surface and lower atmospheric temperatures are complex (cooling and heating) and ambiguous, and may also be connected to semi-direct effects in relation to the perturbations in cloud cover, which vary by as much as -0.72% on the monthly average. Cloud adjustments are likely to be a lagged response to the presence of dust, for example due to its temperature perturbations, such that the influence of

additional Aralkum dust on the atmospheric environment endures after the dust has been deposited or transported elsewhere. Such adjustments would also have feedbacks on the radiative effects of the dust on the longer (non-instantaneous) timescale. The least ambiguous temperature effects occurred in March 2016, the month which saw the biggest dust events of the simulation year, with temperature responses to Aralkum dust which are cooling both on the ground and in the troposphere at 950 hPa, with monthly mean decreases of up to -0.021 K at the surface and -0.009 K in the troposphere.

The estimate that dust over the Aralkum radiatively cools the atmosphere appears to be unusual in the global context (e.g. Albani et al., 2014). However the sensitivity of the simulations to the absorption properties of the dust shows that the overall radiative effect in the atmosphere may be cooling or heating depending on the assumed dust optical properties. It is the case that there remain open questions as to the mineralogy and therefore the optical properties of Aralkum dust, which have not been extensively considered in this paper. It is understood that lakebed dust has a different mineralogy (e.g. Hamzehpour et al.,

2022, and others) to dust from older deserts such as the Karakum, presumably with more strongly scattering optical properties due to the enhanced salt content of the dust (e.g. Argaman et al., 2006; Löw et al., 2013; Indoitu et al., 2015, and others). This would tend to lead to more cooling in the atmosphere. It would in future be worthwhile to explore further the sensitivity of the radiative impacts of the Aralkum dust to the assumed optical properties of this dust. We therefore recommend further investigation of the spectrally-resolved SW and LW optical properties of Aralkum dust, so as to more robustly simulate the

radiative effects of Aralkum dust.

*Code and data availability.* The COSMO model is distributed to research institutions free of charge under an institutional licence issued by the Consortium COSMO and administered by DWD (see http://www.cosmo-model.org/content/consortium/licencing.htm, last access: 10 October 2023). The COSMO licence also includes access to lateral boundary data provided by DWD. ICON analysis data used for the initial and lateral boundary conditions for the COSMO-MUSCAT model experiments in this study can be downloaded from the DWD PAMORE

(Parallel Model data Retrieve from Oracle databases) web-interface (https://www.dwd.de/DE/leistungen/pamore/pamore.html, last access: 10 October 2023). The aerosol-chemistry-transport model MUSCAT is based at least in part on the source code of the COSMO model, and hence redistribution is limited by the COSMO license. The python scripts used to analyse the data and to plot the figures are publicly available on Zenodo (Banks et al., 2023b). The associated COSMO-MUSCAT simulation output data (dust radiative effects, emissions, DODs, and associated atmospheric variables) are also publicly available on Zenodo (Banks et al., 2023a), for the four dust scenarios analysed

in this paper: DUBLT_PRESENT, DUBLT_PAST, DUBLT_NODUST, and DUBLT_ABS. The Global Surface Water dataset (Global Surface Water, 2016), which is used to distinguish the Past and Present DUBLT scenarios, is provided by the European Commission's Joint Research Centre, part of the Copernicus Programme. CERES radiative fluxes data are provided by NASA's Langley Research Center (CERES Science Team, Hampton, VA, USA, 2023).

*Author contributions.* JRB performed the COSMO-MUSCAT simulations and the analysis, and designed and wrote the manuscript. BH and KS contributed to the concept, design and editing of the manuscript.

*Competing interests.* The authors declare that they have no conflict of interest.

*Acknowledgements.* JRB has been funded for this work by the Deutsche Forschungsgemeinschaft (DFG), under the project DESERT-TIME (grant number BA 6612/1-1, project number 414044717). Deutscher Wetterdienst (DWD) has provided access to the COSMO model as well as boundary data. CERES data were obtained from the NASA Langley Research Center CERES ordering tool at https://ceres.larc.nasa.gov/data/. We would also like to thank the editor and the two reviewers for their constructive and insightful comments during the review process.

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
