# Peer review of "Dust aerosol from the Aralkum Desert influences the radiation budget and atmospheric dynamics of Central Asia"

_EGUsphere, 2023_

## Referee Comment (RC3)

Comments on the "Radiative cooling and atmospheric perturbation effects of dust aerosol from the Aralkum Desert in Central Asia" by J. R. Banks et al.

With the help of COSMO-MUSCAT model and a series of sensitivity experiments, the authors investigated the DREs and atmospheric perturbations of dust aerosols from the Aralkum Desert. The topic is relevant, the methodology is well established, and the datasets are frequently used in other studies. However, due to the issues listed below, I recommend a major revision before the manuscript is acceptable for publication.

Major issues:

1. The authors found that, on the yearly timescale, the net surface DRE is --1.34±6.19 W m$^{-2}$, of which -0.15±1.19Wm$^{-2}$ is from the Aralkum dust. Moreover, in the atmosphere, the yearly DRE is -0.62±2.91Wm$^{-2}$ and -0.05±0.51Wm$^{-2}$ comes from the Aralkum dust. As you can see, the uncertainties (the authors did not introduce what are the meaning of uncertainties either) are generally larger than the average state by one order of magnitude, making it difficult to determine whether the dust is cooling or warming the land surface/atmosphere.
2. The general finding from this study is that the Aralkum dust is cooling both the land surface and the atmosphere on the yearly timescale. Many previous studies argued that dust aerosols heat the atmosphere, at least in the aerosol layer. The findings in this study are new, however, the authors did not provide a clear explanation for their novelty.
3. The English in the manuscript needs significant improvement. There are many instances of English misuse, particularly with numerous long sentences and dependent clauses, making comprehension very difficult.

Minor points:

1. L66: The section title is too long, think about shortening it.
2. L74: Could you show DUBLT domain also in Figure 1, as it was mentioned multiple time in the manuscript and is supposed to be important to understand the results.
3. L172: In Figure 3a, the author are comparing simulated DODs to MODIS AODs, which are two different variables.
4. Figure 3: What are the black and green points in Figure 3 a, c, e, and g? The black points in Figure 3a should not be CERES. What is the RMSD? The Bias and RMSD should have a unit.
5. L181: "7.51 and 5.22 Tg", from which plot those two values are obtained.
6. Figure 4: You need to indicate the black and cray points as a legend in the plots so that audience can clearly notice the differences.

7. L213: What is "SKT"?
8. L226: panel (i) -> Figure 6i
9. L227: panel (j) -> Figure 6j
10. Figure 6: Suggest removing the black grid lines or using a light color, which obscure the shading plots. Also for Figure 7 and Figure 10.
11. Figure 7: I am completely confused by this plot. For example, what are panels a to c, and what are panels d to f. Is the colorbar "present DOD at 550 nm" for (a) to (l), while colorbar "present-past DOD difference" is for (m) to (x)?
12. L268: "0.25-0.32 ($10^{-0.6}$-$10^{-0.5}$)", which is not understandable.
13. L403: "as much as -0.72%", is this significant?

---

## Author Response (AR1)

**Author response**

We thank the reviewers for their very helpful comments, and have made appropriate changes to the manuscript. In the collated responses below, the reviewers' comments are included in bold, with our responses included underneath point-by-point.

**RC1/RC2:**

**General comments**

**Using COSMO-MUSCAT model, the manuscript quantified the DRE of Aralkum dust, and investigates the climate perturbations to the atmospheric environment from Aralkum dusts. Their results are interesting for evaluation of radiative forcing of regional dusts (dust emitted by the Aralkum) and understanding the climate perturbations. Several points of the manuscript still need to be improved before accepted. Specifically, Aralkum dust is cooling both at the surface and in the atmosphere, which is different from many previous studies for cooling the atmosphere. Therefore, the manuscript needs to make major revisions before their paper is considered acceptable. Please see the following comments.**

**Main comments**

**1, in the Abstract, the yearly mean net surface DRE is -1.34±6.19 W m$^{-2}$, the mean value is -1.34 W m$^{-2}$, what is meaning of standard deviations for ±6.19? Is it for daily, monthly, or year? The authors should claim the meaning of standard deviations. If the values represent the monthly variations, it is interesting to show the monthly variations of DRE (e.g., which month for largest and smallest value) and the corresponding dust burden or DOD.**

The standard deviations are the standard deviations on the yearly mean. The calculation is a two-step process: firstly for each timeslot an area-averaged value is calculated over the Aralkum box; and secondly a yearly mean and standard deviation are calculated on the timeseries containing ndays × nhours timeslots. This is now stated more explicitly in the caption to Table 1. We also agree that the monthly variations are interesting and relevant, so we introduce a Table 3 to quantify these values, complementing the boxplots in Figure 9.

**2, The authors claimed that in the atmosphere the yearly mean DRE is -0.62±2.91 W m$^{-2}$, of which -0.05±0.51 W m$^{-2}$ comes from Aralkum dust: on the yearly timescale Aralkum dust is cooling both at the surface and in the atmosphere. However, many previous studies show that dust aerosols can warm the atmosphere effectively at the dust layer (Miller et al., 2014; Albani et al., 2014; Scanza et al., 2015; Xie et al., 2018). I think the authors should focus on analyzing these differences between these two results and summarized these reasons.**

We have added an extra subsection (4.2) to Section 4 in order to discuss this slightly unusual behaviour, setting it into the suggested literature context. The new Table 4 quantifies these comparisons. The dust simulated in this paper is of a particularly optically scattering nature, which we argue is a reasonable assumption because of the prominence of salt dust in the region (e.g. Hofer et al., 2020), in contrast to the global view where dust may be expected to be more absorbing. As a sensitivity study we also present results from a more absorbing dust scenario within the DUBLT modelling system, which results in an atmospheric warming of +0.96 W m$^{-2}$ and which is consistent with the results from the Community Atmosphere Model papers. The reason that the scattering dust results in an overall atmospheric cooling effect is that the daytime atmospheric heating is relatively weak compared to the night-time cooling effect, with the night-time cooling mode lasting longer into the period when the sun is low above the horizon (Figure 8). The absorbing dust model run (DUBLT_ABS) has been added to the published Zenodo dataset.

With the inclusion of this discussion, making use of the DUBLT_ABS scenario, we would like to rename the paper "Dust aerosol from the Aralkum Desert influences the radiation budget and atmospheric dynamics of Central Asia".

**3, The authors also show that dust aerosols decrease the cloud cover through semi-direct effects in the atmosphere. It is noted that dust aerosols indirectly act as ice nucleating particles to increase the ice or mixed-phase clouds and**

affect global and regional climate (e.g. DeMott et al., 2010; Tan et al., 2016). The author should add the corresponding content.

An extra sentence on this subject has been added to the final paragraph of Section 5, with the corresponding references included.

**RC3**

With the help of COSMO-MUSCAT model and a series of sensitivity experiments, the authors investigated the DREs and atmospheric perturbations of dust aerosols from the Aralkum Desert. The topic is relevant, the methodology is well established, and the datasets are frequently used in other studies. However, due to the issues listed below, I recommend a major revision before the manuscript is acceptable for publication.

**Major issues:**

1. The authors found that, on the yearly timescale, the net surface DRE is -1.34$\pm$6.19 W m$^{-2}$, of which -0.15$\pm$1.19 W m$^{-2}$ is from the Aralkum dust. Moreover, in the atmosphere, the yearly DRE is -0.62$\pm$2.91 W m$^{-2}$ and -0.05$\pm$0.51 W m$^{-2}$ comes from the Aralkum dust. As you can see, the uncertainties (the authors did not introduce what are the meaning of uncertainties either) are generally larger than the average state by one order of magnitude, making it difficult to determine whether the dust is cooling or warming the land surface/atmosphere.

The +/- notation here refers to the standard deviations associated with the mean values. This is now stated more clearly in the abstract, the results, and in the conclusions. Given the variability in these values, it is indeed the case that dust events occurring at different times and seasons may be causing opposite (heating or cooling) effects to each other: this in fact is one of the messages of this paper. We would particularly like to draw attention to Figure 8, which shows that the major dust events are causing atmospheric warming during the summer (when the sun is high in the sky), but cooling during the winter (when the sun is low, and close to the horizon).

2. The general finding from this study is that the Aralkum dust is cooling both the land surface and the atmosphere on the yearly timescale. Many previous studies argued that dust aerosols heat the atmosphere, at least in the aerosol layer. The findings in this study are new, however, the authors did not provide a clear explanation for their novelty.

On the global scale it is indeed unusual that the dust cools the atmosphere on the yearly timescale, with a more typically understood atmospheric warming effect. The results over the Aralkum are a consequence of the relatively reflecting (i.e. scattering) dust type that has been chosen for this study. It is very important to note that dust in Central Asia is now understood to have an unusually high salt content (e.g. Fomba et al., 2019; Hofer et al., 2020; Xi, 2023), and that it is reasonable to consider Central Asian dust as being a more optically scattering type compared to dust from the Sahara, for example. Because of these scattering properties, the dust is relatively weakly warming during the daytime, and the daytime warming period is shorter than the night-time cooling period. This may seem like an unsatisfyingly ambiguous story, and perhaps contrary to a global perspective, but it is a potential consequence of (scattering) dust events occurring in a mid-latitude region ($\sim$45°N) during different seasons of the year. We have added a Section 4.2 in order to discuss these findings in more detail, in comparison with values from previous studies.

3. The English in the manuscript needs significant improvement. There are many instances of English misuse, particularly with numerous long sentences and dependent clauses, making comprehension very difficult.

We recognise that in places the language became contorted while trying to explain the concepts. This appears to have been most problematic in Sections 4 and 5. We have gone through the manuscript in order to simplify the language, splitting up numerous sentences for clarity.

**Minor points:**

1. L66: The section title is too long, think about shortening it.

This has been amended to "The COSMO-MUSCAT model and its treatment of dust radiative effects".

**2. L74: Could you show DUBLT domain also in Figure 1, as it was mentioned multiple time in the manuscript and is supposed to be important to understand the results.**

The DUBLT domain has been added as panel (b) to Figure 1, marking both the DUBLT domain and the Central Asian region from panel (a).

**3. L172: In Figure 3a, the author are comparing simulated DODs to MODIS AODs, which are two different variables.**

It is true that DUBLT DODs and MODIS AODs are different variables. However in an attempt to discriminate primarily dust AODs, the MODIS AODs have been filtered by the Ångström coefficient. This information has been added to the caption.

**4. Figure 3: What are the black and green points in Figure 3 a, c, e, and g? The black points in Figure 3a should not be CERES. What is the RMSD? The Bias and RMSD should have a unit.**

The black points in Figure 3(a) are MODIS AODs, we have added legends to all four panels in the left column to clarify which points are from which dataset. The RMSD is the root-mean-square difference, now stated in the caption. Units have also been added to the biases and RMSDs.

**5. L181: "7.51 and 5.22 Tg", from which plot those two values are obtained.**

These all-sky emission values are not plotted, although they relate to the cloud-screened values plotted in Figure 4, panels (a) and (b). To clarify that these are emission values taken from the simulation output data and not explicitly shown, this sentence now starts "However under all-sky conditions (not plotted) these emission values..."

**6. Figure 4: You need to indicate the black and cray points as a legend in the plots so that audience can clearly notice the differences.**

The words "DUBLT_PRESENT" and "DUBLT_PAST" have been added as a legend to Figure 4(a) in black and cyan, similarly as in Figure 3.

**7. L213: What is "SKT"?**

"SKT" is the skin (i.e. surface) temperature. We realise that we have not clearly introduced this abbreviation, so this has been added to this sentence and to the caption to Figure 5.

**8. L226: panel (i) -> Figure 6i**

We have changed this to "Figure 6(i)".

**9. L227: panel (j) -> Figure 6j**

Likewise, we have changed this to "Figure 6(j)".

**10. Figure 6: Suggest removing the black grid lines or using a light color, which obscure the shading plots. Also for Figure 7 and Figure 10.**

Thank you for the suggestion, we have amended these plots to have slightly thinner grid lines. The grid lines are however still important for geolocation, so they should remain in the plots.

**11. Figure 7: I am completely confused by this plot. For example, what are panels a to c, and what are panels d to f. Is the colorbar "present DOD at 550 nm" for (a) to (l), while colorbar "present-past DOD difference" is for (m) to (x)?**

The "present DOD" colorbar is for panels (a-c), (g-i), (m-o), and (s-u). We try to clarify this by adding to the caption the sentence: "Panels (a-c), (g-i), (m-o), and (s-u) represent the DUBLT_PRESENT DODs, with the corresponding monthly mean DUBLT_PRESENT - DUBLT_PAST DOD differences plotted in the panels directly beneath." The three green lines were originally included in this Figure in an attempt to delineate clearly the sets of months. The intention of this layout is that the overall DODs are placed side-by-side with the DOD differences, so that the significance of the extra Aralkum dust can be set into the

context of the overall regional dust loading. For some months (e.g. November, December, and March) the extra Aralkum dust is a major player in the Central Asian dust loading, while in other months (e.g. April) it is very minor.

**12. L268: "0.25-0.32 ($10^{-0.6}$-10-$^{0.5}$)", which is not understandable.**

In parentheses are the logarithmic values, i.e. $10^{-0.6} = 0.25$. To try to make this more clear, the part in parentheses is now written: "(understood logarithmically as the range from $10^{-0.6}$-$10^{-0.5}$)".

**13. L403: "as much as -0.72%", is this significant?**

Indeed this value is not very substantial, we agree. We had also written "these appear to be modest..." on lines 396-397 and "The least ambiguous temperature effects..." on line 403. Earlier in the sentence which includes the phrase "as much as -0.72%" we now include the words "... 
[revised manuscript text omitted]

---

## Author Response (AR2)

**Author response**

**Anonymous reviewer 1**

**Thanks very much for your responses. I am ok with the revised version. I have only one comment: in line 40, you should add the corresponding references with dust radiative forcing depending on the altitude of the dust within the atmosphere, the time-of-day, the season, the surface albedo, and the precise mineralogy and optical properties of the dust.**

Many thanks for your comment. We have added several references to support this statement in the introduction.

**Anonymous reviewer 2**

**Most of my concerns proposed in the first round have been addressed, and the authors have revised the manuscript accordingly. However, some issues still need to be addressed before it is acceptable for publication.**

**Firstly, the main conclusion of this study is that dust aerosol from the Aralkum desert has a cooling effect both at the surface and in the upper atmosphere. However, the uncertainty (standard deviation of DREs) is larger than the average value by one order of magnitude, which makes the conclusion less convincing. The authors revised the manuscript and added section 4.2 to demonstrate that the contrast to other findings is due to the scattering property of the dust aerosol from the Aralkum. However, the authors still need to analyze and discuss the uncertainty in a more decent way.**

It is true that the standard deviations on the average DREs are typically larger than the mean values, although not quite as much as an order of magnitude. However we would argue that these means contain both positive and negative numbers, dragging the mean values closer to zero than the standard deviations.

In response to the editors' comments, we have added more discussion on the SW and LW components of the overall DREs, explaining the contributions of these to the DRE patterns which underpin the overall numbers. We have also added quantitative information on the TOA DREs, and further discussed probable lagged responses to the DREs, which may not be accounted for in the instantaneous DREs quoted in the manuscript.

**Secondly, the quality of multiple figures is out of standard. I proposed some issues in the first round, but the authors did not make any revisions. The black grid lines need to be removed in Figures 2,5,6,7,10, which do not help for the geolocation as there are already lat/lon values in the x/y-axis and national boundary lines. Moreover, the text size in many figures is too small to read and needs to be improved.**

We concede that the grid lines did make these maps a bit too heavily cluttered, and so we have removed them, making use of the lat/lon ticks and the country borders for geolocation. We have also added the Aralkum box to more of these maps, to define the study region.

Meanwhile we have also increased the font sizes in many of the figures and we hope that this makes them more legible.

**Editor's report**:
**The atmospheric cooling needs to be better justified. A separation on shortwave and longwave contributions on the atmospheric cooling would be useful.**

We thank the editor for this comment, and recognise that adding more information on the SW and LW contributions is important. To this end we have added an extra figure at the beginning of Section 4.1 to depict the SW and LW contributions to the DUBLT_PRESENT SFC and ATM DREs plotted in Fig. 8 of the previous version of the manuscript (in relation to the DOD, SZA, and season). This new figure is now Fig. 8, while the previous Fig. 8 becomes Fig. 9. Meanwhile Table 1 has also been expanded to include the SW and the LW contributions alongside the net values.

In addition, at the end of p. 20, the following sentences have been included in relation to Table 1:
"The competing SW heating and LW cooling effects on the atmosphere result in a net cooling of the atmosphere to a value of -0.62±2.9 W m$^{-2}$ on the yearly timescale (j), with daytime SW heating of the atmosphere of +3.21 W m$^{-2}$ (k) being outweighed by the cumulative daytime and nighttime LW cooling (l). During the daytime there is a net heating of the atmosphere by +1.24 W m$^{-2}$ due to the greater intensity of the SW effect compared to that of the LW."

Furthermore, we thank the editor for additional helpful comments that he has made during the review process. To this end, we have included the TOA values in Table 1 and we have also added an extra paragraph on p. 21 to comment on the likely consequences of lagged radiative effects and associated temperature adjustments, which are not explicitly considered on the instantaneous timescale. These are also discussed further in the fourth paragraph of the conclusions.